# Cannabinoid Receptors: An Update on Cell Signaling, Pathophysiological Roles and Therapeutic Opportunities in Neurological, Cardiovascular, and Inflammatory Diseases

**DOI:** 10.3390/ijms21207693

**Published:** 2020-10-17

**Authors:** Dhanush Haspula, Michelle A. Clark

**Affiliations:** 1Molecular Signaling Section, Laboratory of Bioorganic Chemistry, National Institute of Diabetes and Digestive and Kidney Diseases, Bethesda, MD 20892, USA; dhanush.haspulagiridhar@nih.gov; 2Department of Pharmaceutical Sciences, College of Pharmacy, Nova Southeastern University, Fort Lauderdale, FL 33314, USA

**Keywords:** endocannabinoid system, neurological diseases, cardiac functions, metabolic diseases, immune functions

## Abstract

The identification of the human cannabinoid receptors and their roles in health and disease, has been one of the most significant biochemical and pharmacological advancements to have occurred in the past few decades. In spite of the major strides made in furthering endocannabinoid research, therapeutic exploitation of the endocannabinoid system has often been a challenging task. An impaired endocannabinoid tone often manifests as changes in expression and/or functions of type 1 and/or type 2 cannabinoid receptors. It becomes important to understand how alterations in cannabinoid receptor cellular signaling can lead to disruptions in major physiological and biological functions, as they are often associated with the pathogenesis of several neurological, cardiovascular, metabolic, and inflammatory diseases. This review focusses mostly on the pathophysiological roles of type 1 and type 2 cannabinoid receptors, and it attempts to integrate both cellular and physiological functions of the cannabinoid receptors. Apart from an updated review of pre-clinical and clinical studies, the adequacy/inadequacy of cannabinoid-based therapeutics in various pathological conditions is also highlighted. Finally, alternative strategies to modulate endocannabinoid tone, and future directions are also emphasized.

## 1. Introduction

### Discovery of the Endocannabinoid System

The therapeutic potential of cannabis, commonly known as marijuana, has been a subject of great interest for several centuries. Its anxiolytic and euphoric properties were acknowledged in religious scriptures that date back to several millennia [1]. Many cultures and civilizations have used cannabis preparations to treat a variety of ailments, ranging from rheumatism, inflammation and malaria for several millennia [2]. While evidence of therapeutic utility of cannabis was known in Asia and Africa, cannabis was relatively unknown to the western world until the 19th century [3]. The first scientific report on cannabis was published by the Irish physician, William ‘O Shaughnessy, which marked the earliest traces of cannabis globalization. By providing evidence of its therapeutic efficacy and safety for pathological conditions such as infantile convulsions and cholera, he was instrumental in laying the foundation for cannabis research [3]. Pioneering works from the groups of Todd, Adams, and Mechoulam, in the 20th century, led to a better understanding of the chemical makeup of cannabis [4,5,6,7].

The first report of the existence of the brain cannabinoid receptor, termed the cannabinoid receptor type 1 (CB1R), was reported by Howlett’s group in the late 1980s [8]. The discovery of the CB1R was followed by the identification of the second cannabinoid receptor, termed the peripheral cannabinoid receptor (due to the lack of expression in brain) or cannabinoid receptor type 2 (CB2R) by Abu-Shaar’s group [9], and their two endogenous ligands, anandamide and 2-arachidonoylglycerol (2-AG) [10,11,12], by Mechoulam’s and Waku’s groups. The identification of biosynthetic and degradative pathways of endocannabinoids in the following years, make up the classical endocannabinoid system [13,14,15,16,17]. These seminal discoveries laid the foundations for endocannabinoid research. For a detailed account on cannabinoid and endocannabinoid history, please refer to the excellent reviews by Di Marzo [3] and Pertwee [18].

## 2. Cannabinoid Receptors

### 2.1. Molecular Architecture

The CB1 gene ***(****CNR1),* which was first cloned by Matsuda et al. [19], encodes for the CB1R. The human *CNR1* is located on chromosome 6 (6q15, HGNC:2159) [20], and comprises of four exons, with exon 4 described as the main coding exon [21,22]. The mouse and rat *CNR1*, localized on chromosome 4 and 5 respectively, comprises of a single promoter [23]. The CB1R, a 53 kDa protein, is glycosylated post-translationally resulting in a 64 kDa glycosylated form [24]. The glycosylated fraction supersedes the non-glycosylated fraction in abundance [24]. The CB1R belongs to the Class A G protein coupled receptor (GPCR) subfamily, often regarded as one the most diverse subfamilies of GPCRs in humans [25]. Similar to the other GPCRs in this family, the CB1R comprises of a seven-transmembrane helical domain, extracellular and intracellular loops, an extracellular N terminus and an intracellular carboxy terminal tail [25,26]. The extracellular N terminal region is an important factor in conferring receptor stability and trafficking [27,28]. Although the N-terminal region has been reported to have a minor role in orthosteric binding, the membrane-proximal region of the tail is implicated in the modulation of the binding affinity of allosteric ligands [27,29]. Published crystal structure studies of inverse agonist- and antagonist-bound CB1R further helped to characterize the ligand binding pocket and provided insight into the importance of the extracellular loop, the N-terminal region, and the specific transmembrane helices for docking of agonists [30,31]. Similar to the other GPCRs, the second and the third intracellular loops have been demonstrated to have an important role in both receptor activation and desensitization [32,33], while the carboxy terminal tail mostly regulates desensitization and internal sorting [34,35]. Phosphorylation of serine and threonine residues located in the carboxy terminal region has been shown to regulate receptor desensitization and internalization via interaction with the CB1R interacting protein (CR1P) 1a [36], G protein coupled receptor kinases [34] and arrestins [37]. Interestingly, the carboxy terminal region has also been reported to adopt two amphipathic helical domains, which are functionally capable of affecting receptor signaling, polarity and surface expression [38,39,40,41].

The CB2 gene *(CNR2),* which was first cloned by Munro et al. [9], encodes for the CB2R. The human *CNR2* is located on chromosome 1 (1p36.11, HGNC ID: 2160). Unlike *CNR1*, both human and mouse *CNR2* have been reported to comprise of two separate promoters [42]. The CB2R has a similar structure to the other GPCRs in this class. It comprises of 7 transmembrane domains, N terminus and C terminus, 3 extracellular and intracellular loops, and also an amphipathic cytoplasmic helix [43,44,45]. The CB2R shares 44% of overall sequence identity, and 68% of transmembrane sequence identity, although the sequence identity has been reported to be lower in TM1, TM4 and TM5 [9,46]. Unlike the CB1R, the CB2R does not have a long N-terminal region. Other key differences include an aromatic-rich environment in the TM5 of CB2R, and a lack of phosphorylation site for PKC in the third intracellular loop in CB2R, the latter of which is present in CB1R [46]. The second intracellular loop in combination with the carboxy terminal region plays a pivotal role in CB2R-mediated signal transduction [47].

### 2.2. Ligands

#### 2.2.1. Endogenous Cannabinoids

Both CB1Rs and CB2Rs are class A, lipid-like GPCRs that are activated by endogenously produced lipophilic ligands [48]. The prototypical endogenous cannabinoids or endocannabinoids are 2-AG and anandamide. 2-AG and anandamide are eicosanoids that are synthesized on-demand from arachidonic acid-containing phospholipids, such as phosphatidylinositol 4,5-bisphosphate (PIP2) and phosphatidylethanolamine (PE), respectively. These ligands have complementary as well as divergent functions [49]. While 2-AG is a full agonist at both CB1Rs and CB2Rs, anandamide is a partial agonist for both receptors. Other lesser-known endocannabinoids or non-classical eicosanoids include, N-acyl dopamine (NADA) and 2-arachidonyl glyceryl ether (noladin ether), both of which bind strongly to the CB1R [50,51]. Additionally, virodhamine was identified to be a full agonist at the CB2R, and have antagonistic activity at the CB1R [52]. Apart from endogenous orthosteric ligands, endogenous allosteric modulators for the CB1R and the CB2R have also been identified. Lipoxin A4 and pepcan 12 are both reported to be positive allosteric modulators of CB1Rs and CB2Rs, respectively [53,54].

#### 2.2.2. Exogenous Cannabinoids

Exogenous cannabinoids comprise of both naturally occurring phytocannabinoids, such as Δ^9^- tetrahydrocannabinol (THC), and synthetic cannabinoids. THC has a high affinity for both the CB1R and the CB2R. Synthetic cannabinoids such as HU-210, R-()-WIN55212 and CP55940 also display high affinity for both receptors. ACEA, noladin ether, and arachidonylcyclopropylamide display higher affinity for the CB1R when compared to the CB2R, while JWH-133, HU-308, and JWH-133 display higher affinity for the CB2R when compared to the CB1R [55]. For the classification based on chemical structure, please refer to the report by the International Union of Basic and Clinical Pharmacology (IUPHAR) [55]. The various components of the endocannabinoid signaling system, along with endogenous cannabinoid modulators and the exogenous cannabinoid receptor ligands (phytocannabinoids and synthetic cannabinoids) are shown in Figure 1. While this review focusses on mostly the CB1R and the CB2R, it is important to understand that endogenous and exogenous cannabinoids are also capable of interacting with an array of different receptors, channels, and proteins [56,57].

### 2.3. Tissue and Cellular Distribution

Cells of the central and the peripheral nervous system: The CB1R is highly expressed in most regions of the central nervous system (CNS), with densities that rival other neurotransmitter and neuromodulatory receptors [58]. Moderate to high expression of CB1Rs have been observed in the cerebral cortex (cingulate gyrus, prefrontal cortex, and hippocampus), basal ganglia (globus pallidus, substantia nigra), periaqueductal gray, hypothalamus, amygdala, and cerebellum [58]. The CB1R expression is relatively sparse in the medullary respiratory control centers of the brainstem. However, the CB1R is highly expressed in brainstem medullary nuclei, such as the nucleus of the solitary tract and area postrema, serving as the primary integrative centers for the cardiovascular system and emesis, respectively. Moderate CB1R expression was also confirmed in the spinal cord (dorsal horn and lamina I, III, and X) [58,59,60]. More recently, dense CB1R positive fibers were identified in the ventral horn [61]. Apart from the CNS, CB1R expression was reported in the somatic, sympathetic, parasympathetic, and the enteric nervous systems [61,62,63,64,65,66,67,68,69,70].

Neuroanatomical studies identified the CB1R to be primarily located in the presynaptic terminals of GABAergic (amygdala and cerebellum) [71], glutamatergic (cortex, hippocampus and amygdala) [71,72,73], dopaminergic [71], GABAergic interneurons [74,75], cholinergic neurons [76], noradrenergic [77], and serotonergic neurons [78]. Low levels of presynaptic CB1R were also detected in the nociceptive primary afferent fibers in the spinal cord [79]. In addition to neuronal cells, CB1Rs were also identified in the perivascular and perisynaptic astroglial processes [72,80] and oligodendrocytes [81]. Apart from astrocytes, the CB1R was also identified in other cells of the blood brain barrier (BBB) such as the brain endothelial cells [82], pericytes, and vascular cells [83]. Apart from the glial cells in the CNS, the CB1R was also observed in myelinating Schwann cells of the peripheral nervous system [61]. The CB2R on the other hand is expressed at very low levels in the CNS under physiological conditions. However, pathological conditions characterized by a neuroinflammatory state resulted in an upregulation of CB2R levels in glial cells, such as microglia [84,85,86,87].

Non-neuronal cells: The CB1R has also been reported in the peripheral organ systems of the body, albeit at lower, but functional levels [88,89]. Functional CB1R have been reported in the liver [90], muscle [91], adipose tissue [92], vasculature [93,94], heart [95,96], pancreatic beta cells [97], reproductive organs [98], and alveolar cells [99]. The CB2R however is expressed in high levels in immune cells and in lymphoid tissues. Cells that participate in both innate and adaptive immune response, such as the spleen, thymus, and peripheral blood mononuclear cells, are known to express high levels of CB2R [45,100]. Interestingly, under pathological conditions, various peripheral cell types have also been shown to express detectable levels of the CB2R. This includes activated hepatic stellate cells [101] and renal cells from fibrotic kidney [102], and lung tissue from rats subjected to chemical-induced lung injury [103]. However, there have been reports of detectable levels of the CB2R under normal physiological conditions in pancreatic acinar cells [104], adipocytes [105], skeletal muscle cells [106], cardiomyocytes, and endothelial cells [107]. Additionally, both CB1R and CB2R have been detected in connective tissue such as fascial fibroblasts and osteoclast cells [108,109]. An overview of tissue distribution of the CB1R and the CB2R in healthy conditions is shown in Figure 2.

### 2.4. Signaling

#### 2.4.1. Canonical

Inhibitory G protein-coupled and β-arrestin signaling: Classically, the CB1R couples to the pertussis toxin (PTX)-sensitive G-protein (Gα_i/o_), which results in the inhibition of forskolin-stimulated cAMP, activation of G-protein coupled inwardly rectifying potassium channels (GIRKs), and an inhibition of several calcium channels [110,111,112,113]. G protein-mediated signaling in the CB1R is followed by the recruitment of β-arrestins 1 and 2, which play a role in receptor internalization and signaling [114,115]. G-protein-coupled receptor kinases along with β-arrestin 2 have been shown to trigger receptor desensitization, internalization, and G-protein signal termination, while β-arrestin 1 is involved in activation of mitogen-activated protein kinases (MAPKs) and also regulation of gene expression [116,117,118,119]. Both β-arrestin 1 and 2 have also been implicated in ERK1/2 activation [120]. MAPKs such as ERK1/2, p38, and c-Jun N-terminal kinase (JNK) were shown to be activated in response to CB1R agonists in a variety of cell types [121,122,123]. Additionally, the CB1R activation was also shown to activate the PI3K/AKT pathway resulting in the regulation of neuronal survival [124]. Activation of MAPK, JNK, and PI3K/AKT pathways via the CB1R have all been linked to induction of several transcription factors such as Krox-24 [121], CREBH [122], and BDNF [124]. For a detailed review on the CB1R cell signaling, please refer to the review by Turu and Hunyady [125].

Similar to the CB1R, the CB2R is also a GPCR that couples to the inhibitory G-protein. Stimulation of CB2R has also been shown to activate MAPKs and subsequent induction of Krox-24 expression [126]. Additionally, the CB2R has also been shown to promote neuronal survival through the activation of the PI3K/AKT and JNK pathway [127]. Also, CB2R activation results in significant β-arrestin 2 recruitment and β-arrestin 1-mediated MAPK activation [128,129].

#### 2.4.2. Non-Canonical

Stimulatory G protein coupled signaling: While the CB1R has been reported to preferentially couple to Gα_i/o_ type G proteins, some endogenous and synthetic cannabinoids, have been observed to enhance CB1R-mediated coupling to stimulatory G proteins such as Gα_q/11_ and Gαs [130,131]. Interestingly, stimulated CB1R was shown to access two distinct G protein pools, resulting in biphasic ERK 1/2 activation in primary neuronal cultures [132]. Additionally, the exogenous synthetic cannabinoid WIN55,212-2, has also been reported to enhance calcium levels via a CB1R-Gq-PLC pathway in both HEK293 cells and in cultured hippocampal neurons [130]. Furthermore, endocannabinoids have also been observed to elevate calcium levels and glutamate release via activation of astrocytic CB1Rs [133]. As a result, calcium spikes in response to astroglial CB1R stimulation can have profound consequences on the modulation of synaptic strength. The role of astroglial CB1R in the regulation of synaptic activity is described in a later section. A biphasic ERK 1/2 activation [123], and an elevation in calcium levels via a Gi independent pathway [134], were also observed in other cell types.

Similar to the CB1R, there have been reports on the ability of the CB2R to elevate calcium levels via a PLC pathway in endothelial cells [135], and insulinoma beta-cells [134], suggesting a Gi-independent coupling route. Intracellular administration of CB2R agonists, in CB2R-transfected U2OS cells, triggered dose-dependent activation of calcium responses which was Gq-mediated [136].

Crosstalk: Evidence of crosstalk between the CB1R and other GPCRs was identified by multiple studies. Co-stimulation of the CB1R and dopamine type 2 receptor (D2R) resulted in extensive colocalization, dimerization, and an increase in cAMP levels, suggesting a potential change in signal transduction profile [137]. Heterodimers between CB1Rs and D2Rs, independent of receptor occupancy, was also observed in transiently transfected cells [138]. Further evidence for an interplay between CB1Rs and D2Rs was demonstrated in striatal neurons, whereby a D2R antagonist was able to hinder CB1R agonist-mediated phosphorylation of ERK1/2 [139]. Additionally, the existence of dimers between the CB1R and the Adenosine (A_2A_) receptors [140], and also heterooligomers with CB1R- A_2A_-D2R [141], were also reported in transfected cells. At a functional level, blockade of A_2A_ receptors was shown to attenuate the CB1R-dependent motor depressant effects, suggesting possible interdependency between the two receptors [140]. Similar functional interdependency was observed between the CB1Rs and the orexin receptors, whereby their co-expression in a heterologous cell line resulted in potentiation of orexin-mediated MAPK activity [142]. However, functional antagonism between the CB1R and other GPCRs has also been reported. Co-stimulation of the CB1R and the μ opioid receptors resulted in attenuation of STAT3 phosphorylation and neuritogenesis in Neuro-2A cells [143]. Reciprocal inhibition has also been reported in the hippocampus between CB1Rs and GABA-b receptors [144]. This type of reciprocal inhibition was also observed in astrocytes from our laboratory, whereby co-stimulation of astrocytes with Angiotensin (Ang) II and the CB1R agonist, resulted in significantly reduced MAPK activation when compared to Ang II alone [123]. However, the results presented in this study differ from another where potential heterodimerization between AT1R and CB1R was observed [145]. In this study, co-treatment of Ang II with the CB1R agonist, HU-210, led to an increase in AT1R-mediated activation of ERK1/2 in a neuroblastoma cell line, suggestive of a possible synergism between the two receptors [145]. In the brain, there is evidence of CB1R involvement in AT1R-mediated elevation in blood pressure. An increase in mean arterial pressure from the administration of Ang II into paraventricular nucleus (PVN) of Wistar Kyoto (WKY) rats was blunted by simultaneous infusion of the CB1R inverse agonist, AM251 [146]. This however is indicative of a potentiation of AT1R-mediated effects by CB1R in the CNS. A detailed understanding of the molecular signaling of central and peripheral AT1Rs in pathophysiology and possible crosstalk with CB1R, is beyond the scope of this review. Please refer to reviews from Haspula and Clark [147] and Forrester et al. [148] for a better understanding of the AT1R signaling mechanisms.

The CB2R has also been reported to heterodimerize with the CB1R in co-transfected neuronal cells, and co-stimulation was demonstrated to result in negative AKT phosphorylation and neurite outgrowth [149]. They also exhibited cross-antagonism, whereby CB2R antagonists were demonstrated to block the CB1R-mediated effects, and vice-versa [149]. Additionally, the CB2R has been reported to crosstalk with other receptors. The CB2R-selective ligand, SR 144528, was demonstrated to inhibit downstream signaling of the insulin and lysophosphatidic acid receptors in Chinese hamster ovary (CHO) cells transfected with recombinant CB2R [150]. Additionally, evidence of the CB2R heterodimerizing with the C-X-C Motif Chemokine Receptor 4 (CXCR4) has been observed in metastatic cancer cell lines, resulting in diminished functions of CXCR4 receptor and limited tumor cell migration and progression [151,152].

Transactivation: The interaction between the CB1R and other GPCRs is not limited to receptor dimerization. The concept of paracrine transactivation of the CB1R by Gq GPCRs, was identified in non-neuronal cells by the Hunyady group. In CHO and other commercial cell lines, the CB1R was demonstrated to be transactivated, both in an autocrine and paracrine fashion, by Ang II via the AT1R-Gq-DAGL axis [153,154]. Interestingly, the CB1R could also be transactivated by other Gq GPCRs as well [154]. Studies from the same group also showed an augmentation of AT1R-mediated vasoconstriction by Ang II in the presence of a CB1R antagonist [89]. Interestingly, cannabinoids have also been shown to transactivate other receptors, such as epidermal growth factor receptor (EGFR) and tropomyosin receptor kinase B, resulting in cancer cell proliferation [155], interneuron migration and cortical development [156], and corneal epithelial cell proliferation [157]. For a comprehensive understanding of paracrine transactivation of CB1R by other GPCRs, please refer to the review by Gyombolai [146].

Biased signaling: Biased signaling refers to the propensity of a ligand to preferentially activate a distinct signaling pathway over others. This was observed for both exogenous and endogenous cannabinoids at multiple downstream steps following CB1R stimulation; that is at the level of G protein signaling, cAMP signaling, MAPK activation, and calcium mobilization. For instance, anandamide demonstrated a greater ability to inhibit cAMP and activate ERK1/2 signaling when compared to 2-AG [158]. The endogenous cannabinoid N-arachidonoyl dopamine, which acts as an agonist for both the CB1R and the transient receptor potential cation channel subfamily V member 1 (TRPV1) [159], displayed high functional selectivity for calcium mobilization via Gq protein signaling, while remaining relatively ineffective at activating other canonical signaling pathways [160]. Exogenous cannabinoids such as CP55940, THC, and HU-210 displayed a greater ability, when compared to WIN55,212-2, to inhibit cAMP over ERK1/2 activation [158]. The ability of both endogenous and exogenous cannabinoids to activate distinct signaling pathways and cellular processes could also be attributed to a selection bias for either G-protein or β-arrestin-induced signaling. Neuronal CB1R stimulation by exogenous cannabinoids, THC and CP55,940, triggered significant β-arrestin 2 recruitment, while WIN55,212-2 showed greater preference for Gα_i/o_ and Gβγ signaling [131]. Among the endocannabinoids tested in the same study, 2-AG showed greater preference toward β-arrestin 2 binding, when compared to anandamide [131]. In a cell culture model of Huntington’s disease, 2-AG and AEA demonstrated greater preference to G-protein signaling, while CP55,940 and THC had greater preference for β-arrestin 1 recruitment and reduced CB1R protein levels [161]. Another study observed a high dependence of β-arrestin 1 on the activity of CP55940, but not THC, in both in vitro and in vivo assays [162]. This suggests that functional selectivity of the ligands may be dependent on the cell type and pathophysiological alterations.

Similar to the CB1R, functional selectivity at the CB2R for both endogenous and exogenous cannabinoids was also demonstrated by multiple studies. 2-AG via the CB2R induces a greater degree of ERK1/2 activation, when compared to noladin ether as well as CP55,940 [163]. However, noladin ether and CP55,940 demonstrated greater potency in the inhibition of adenylyl cyclase, when compared to 2-AG [163]. Exogenous cannabinoids such as CP55,940 and WIN55,212-2 exhibited differences with regards to CB2R internalization and inhibition of voltage-gated calcium channels, in spite of ERK1/2 phosphorylation and β-arrestin 2 recruitment [128]. Additionally, in a study profiling functional selectivity of a wide range of CB2R agonists, THC and 2-AG were found to display a strong signaling bias for the CB2R, displaying a significantly greater preference for the activation of their most-preferred pathway [164]. Additionally, SR144528, which is an antagonist for the CB2R, displayed greater efficacy at blocking cAMP signaling when compared to other signal transduction pathways [164]. For a detailed overview of functional selectivity of CB2R signaling, please refer to the review by Dhopeshwarkar and Mackie [165].

Constitutive activity: A constitutively active receptor assumes an active state, initiating downstream signaling, even in the absence of ligand stimulation. High basal signal for the CB1R was not only observed in cells expressing the recombinant form of the receptor, but also seen in cells expressing the native receptor [166,167,168,169,170]. Further evidence supporting this view comes from the subcellular localization of CB1R under baseline conditions. High levels of CB1Rs have been observed to be localized intracellularly in various subcellular compartments, including lysosomes and endosomes [171,172,173]. Considering that a constitutively active CB1R may be involved in maintaining homeostasis, utilizing neutral antagonists over inverse agonists were found to be beneficial in preclinical studies [174,175].

Apart from the CB1R, the CB2R was also reported to exhibit constitutive activity in cells expressing recombinant as well as native receptors [150,176,177]. CB2R has been observed to be located intracellularly in neuronal cells from rat prefrontal cortex [178], and in endolysosomes in CB2R-transfected U2OS cells [136].

Heterologous desensitization: Mackie’s group identified that PKC activators can trigger phosphorylation of CB1Rs at single serine residue (S317) in the third intracellular loop, resulting in the attenuation of receptor activity [179]. Since Gq GPCRs can activate PKC, our laboratory investigated whether other Gq GPCRs, such as the Ang AT1R, could trigger an increase in CB1R phosphorylation at the exact same residue. We observed that Ang II via the AT1R is capable of inducing phosphorylation of CB1Rs at the third intracellular loop, thereby potentially triggering desensitization of the receptor [123]. This was most prominent in the cerebellum [123]. It would be interesting to see if other Gq GPCRs could also trigger phosphorylation of the CB1R via this mechanism.

Intracellular signaling: While both CB1Rs and CB2Rs are predominantly located in the cell membrane, there is evidence of intracellular localization and signaling of CB1R in various organelles such as endocytes, mitochondria, and the nucleus as mentioned previously [171,172,173,180]. Since endocannabinoids are lipophilic, they can traverse the cell membrane and interact with their receptors that are localized on various intracellular organelles and compartments.

Autoinduction: GPCRs usually undergo receptor downregulation in response to persistent activation of the receptor by ligands. The CB1R is similar in this aspect and induction of cannabinoid tolerance is often a biological consequence of exogenous cannabinoid induced CB1R downregulation. However, 2-AG was demonstrated to increase CB1R transcription via CB1R-mediated retinoic acid receptor activation in hepatocytes [181]. Interestingly, acute treatment of both exogenous and endogenous cannabinoids has also been implicated in elevating CB1R mRNA in a variety of cell types and tissues [182,183]. Endocannabinoids released from astrocytes were also elevated by the activation of the astroglial CB1R by cannabinoids [184]. Interestingly, cannabinoids have also been shown to increase the CB1R gene expression via CB2R [185]. An overview of the canonical and the non-canonical signaling is described as a schematic in Figure 3.

## 3. Biological Role of the Cannabinoid Receptors

The previous sections described the molecular architecture of the cannabinoid receptors, their localization and distribution, and their cellular signaling mechanisms. The ubiquitous nature of cannabinoid receptors lends itself to regulate a variety of cellular and physiological processes. From regulation of cellular functions, such as neuromodulation, to orchestrating complex metabolic and immune responses, cannabinoid receptors serve essential role in both physiological and pathological conditions. Since the endocannabinoid system is vital for homeostasis of several biological processes, several pathological conditions pertaining to cardiovascular, neurological, metabolic and immunological diseases, are often associated with alterations of endocannabinoid tone. This section describes the roles of cannabinoid receptors in various biological processes, and the pathological conditions associated with changes in endocannabinoid tone. This includes results from both pre-clinical and clinical studies. For a complete list of recently published clinical studies (2010–2020), pertaining to the aforementioned pathological conditions, please refer to Table 1.

### 3.1. Cellular Morphogenesis and Developmental Roles

#### 3.1.1. Embryogenesis

Embryogenesis refers to the formation of a multicellular embryo from a single cell zygote via series of processes that include division, differentiation, and cell fate specification. Although evidence of endocannabinoid synthesis can be traced back to unicellular organisms [221], the origins of the CB1R closely parallels the evolution of multicellular organisms [222]. This could well be indicative of their importance in cell differentiation and specialization of functions. Both the CB1R and the CB2R were detected prior to the blastocyst stage, and cannabinoid agonists were demonstrated to arrest the growth of an embryo in culture [223]. Elevated levels of the CB1R were also observed in the blastocyst, particularly in the trophectoderm [223,224], and implantation was shown to be mediated by the CB1R and not the CB2R [225]. Furthermore, a greater mRNA expression of CB1R when compared to the CB2R highlights the importance of CB1R signaling in embryogenesis [226]. Additionally, there have been a few reports of a possible involvement of the CB2R during various stages of embryonic development. While both CB1R and CB2R transcripts were identified in the human trophoblast during the first trimester, in vitro experiments demonstrated that anandamide prevented the proliferation of BeWo trophoblast cells via the CB2R [227]. In a recent study using zebrafish embryos, it was demonstrated that blocking CB2Rs during early development resulted in a greater inhibition of heart rate than that observed with a CB1R blocker [228]. Additionally, both CB1Rs and CB2Rs were also reported in embryonic stem cells (ESC) [229,230]. They were also implicated in increasing cell viability in embryoid bodies (EB) [231]. Increased survival of EB serves as a critical preliminary step for ESC differentiation.

#### 3.1.2. Neurodevelopment

Functional CB1Rs were detected in the fetal rat brain as early as gestational day (GD)-14 [232]. Additionally, mRNA transcripts for CB1Rs were also reported in the subventricular zone at GD-21, highlighting its importance in neuronal and glial cell generation in the developing CNS [232]. Apart from critical roles in early neurodevelopment, the CB1R was also shown to regulate neuronal differentiation from embryonic cortical neuron progenitor cells, and regulation of neurogenesis in adult hippocampus [233]. Knockout studies further confirmed the vital role of CB1Rs in adult neurogenesis [234] and neurosphere formation [235]. HU210, a synthetic cannabinoid, was also shown to promote neurogenesis in the hippocampal dentate gyrus [236]. Apart from neurogenesis and neuronal differentiation, the CB1R was also reported to play critical roles in synaptogenesis, pathfinding, and network formation [237,238,239], possibly because of elevated expression in axonal growth cones. The CB1R also regulates neural progenitor cell fate decisions, as evidenced by its involvement in the differentiation of neural progenitor cells toward an astroglial lineage [240]. Interestingly, both CB1Rs and CB2Rs have been shown to regulate oligodendrocyte progenitor cells function, via a PI3K/AKT pathway, whereby their activation exerts a protective role [81,241]. Although the role of CB2R in CNS development is not as well studied, there is evidence of its involvement in neural progenitor cell proliferation, and possibly in neurogenesis [242].

#### 3.1.3. Implications of a Dysregulated Endocannabinoid Tone in Neurodevelopmental Disorders

Cognitive defects associated with developmental abnormalities: Multiple studies have linked alteration in cannabinoid signaling during early developmental stages to various cognitive defects, suggesting a link between defective endocannabinoid signaling and various developmental disorders [243,244,245]. Dysregulated CB1R signaling was reported in a mouse model of fragile X syndrome (*fmr1* knockout), a disorder characterized by intellectual and developmental disabilities [246]. Although another study reported unchanged levels of CB1Rs in mice lacking *fmr1*, an increased glutamate receptor-mediated mobilization of endocannabinoids was reported [247]. Additionally, genetic and pharmacological blockade of CB1Rs in mice lacking *fmr1* was demonstrated to improve cognitive impairment [246,247,248]. Apart from protein expression, *CNR1* was also reported to be significantly downregulated in postmortem brains of individuals with autism [249]. Interestingly, indirect lines of evidence highlight *CNR1* variants to be a key genetic factor in contributing to differences in striatal response to various emotions [250,251].

ADHD: Genetic variants of *CNR1* were also associated with neurobehavioral disorders such as attention deficit hyperactivity disorder (ADHD) [252]. In vitro experiments demonstrated that cannabinoid administration resulted in a reduction in hyperactivity in juvenile spontaneously hypertensive rats (SHRs), a rat model of ADHD [253,254]. Reduced expression of CB1Rs was also observed in the prefrontal cortex of SHRs, suggesting potential hypoactivation of the CB1R [254]. Evidence from case studies also suggests an improvement in ADHD symptoms in response to cannabis and cannabinoids consumption [255,256].

Schizophrenia: Several studies found a strong association of cannabis use to schizophrenia [257,258]. Additionally, evidence of altered emotional status and impulsive behavior in adolescents, due to cannabinoid exposure, has also been reported [259,260]. Intriguingly, by employing radiotracer binding and mRNA assays, altered CB1R expression was also reported in postmortem brains of schizophrenic subjects, that were independent of cannabis use [261,262,263,264]. Quantitative PET imaging techniques also revealed a strong association between changes in CB1R levels in specific brain regions and schizophrenia symptoms [265]. Additionally, in vitro experiments revealed that the temporal changes in CB1R expression in adults coincided with the onset of schizophrenia symptoms, further highlighting the involvement of the endocannabinoid system in schizophrenia [266]. Preclinical studies also established the utility of CB1R blockade as a stand-alone therapy, or in conjunction with an established antipsychotic in the treatment of cognitive defects induced by psychotomimetics in rats [267,268]. Considering that rimonabant, a CB1R antagonist, is discontinued because of neuropsychiatric adverse effects, alternative strategies should be devised to reduce endocannabinoid tone in the treatment of neurodevelopmental disorders such as schizophrenia.

### 3.2. Neurological

#### 3.2.1. Neuromodulation

The ability of the CB1R to fine-tune and regulate GABAergic synaptic transmission via retrograde signaling [269,270,271], known as depolarization induced suppression of inhibition (DSI), remains one of the cornerstones in endocannabinoid research. DSI is the retrograde transmission of messenger signals towards the pre-synaptic cleft resulting in GABAergic synaptic inhibition [272]. Excessive postsynaptic receptor activation results in mobilization of endocannabinoids into the synaptic cleft. They can then traverse the synaptic cleft, from the postsynaptic neuron to the presynaptic neuron, where they activate the CB1R. Activation of CB1Rs attenuates neurotransmitter release into the synaptic cleft, resulting in dampened synaptic activity. This ‘circuit-breaker’ like functionality is a crucial modus operandi of the CB1R by which it influences synaptic plasticity [273]. In addition to dampening the activity of inhibitory neurons, the CB1R is abundantly expressed on presynaptic glutamatergic neurons [71,72]. Hence it can also suppress excitatory neuronal activity [274], thereby affecting the two arms of short-term plasticity. Activation of metabotropic glutamate receptors results in mobilization of endocannabinoids and inhibition of synaptic transmission [275,276]. Additionally, CB1R activation regulates the induction of long-term synaptic depression (LTD) in the nucleus accumbens [277], cerebellum [278], and prefrontal cortex [279]. It also regulates long term potentiation (LTP) in the hippocampus [280]. However, the action of endocannabinoids on LTP is controversial since there has been reports showing that they restrict LTP in the hippocampus [281,282,283], as well as have a dual role on LTP strengthening [284]. CB1R-mediated activation of mTOR and subsequent induction of the protein synthesis machinery is attributed to its modulation of long-term memory [285]. While CB1R-mediated transient response is attributed to inhibiting the activity of calcium channels and activating GIRK, changes in reduction of cAMP/PKA signaling and RIM1α, along with changes in protein synthesis are some of the mechanistic changes linked to a sustained and long term modulation of synaptic activity [273,286,287,288]. For a comprehensive overview of the various mechanisms that govern CB1R-mediated induction of long-term synaptic plasticity, please refer to the review by Heifets and Castillo [289]. CB1R’s widespread distribution in the brain, in conjunction with the ability to affect long term changes in synaptic strengthening, makes the CB1R a critical player in cognition, memory, behavior, and learning.

Relevant to neuronal plasticity, is the ability of astrocytes to communicate with neurons through mobilization of calcium and neurotransmitters [290]. The concept of tripartite synapse first introduced by Araque et al. introduces astroglia as a potential third signaling cell together with the pre- and post-synaptic neuronal cells [291]. Endocannabinoids released from neurons could then activate distant neurons by activating the astroglial CB1R, thereby serving an unidentified role in neuromodulation [133,292]. Endocannabinoid-mediated bidirectional communication between astrocytes and neurons has been demonstrated to significantly impact synaptic plasticity [293,294] and memory formation [295], further underpinning the importance of the astroglial endocannabinoid system in regulating physiological functions that were earlier believed to be exclusively neuronal. This suggests that both the neuronal and the astroglial CB1R could have an important role to play in neuromodulation. For a detailed overview of the molecular mechanisms of the astroglial CB1R in the regulation of short, spike-time-dependent-, and long-term synaptic plasticity, please refer to the reviews by Catillo et al. [296], Navarrete et al. [292], Oliveira da Cruz et al. [297] and Guerra-Gomes et al. [298].

There is conflicting evidence for the existence of the CB2R in the brain. While several studies have identified CB2R expression in both neuronal and microglial cells [299,300], the lack of specificity of several commercially available CB2R antibodies in combination with low CB2R expression in quiescent microglia [85,301,302,303,304], has resulted in inconclusive evidence to prove the existence of the CB2R in healthy brain cells. But by employing knock out mice, several studies revealed the existence of CB2Rs in various regions of the brain [305,306]. Unlike the CB1R, the CB2R was also observed to be located on the post synaptic terminals [307]. Additionally, both glutamatergic and GABAergic neurons in the hippocampus were reported to express CB2R mRNA [304]. Recently, CB2R agonists were reported to reduce dopaminergic neuronal excitability in the ventral tegmental area (VTA) through cAMP reduction [308].

#### 3.2.2. Neuroinflammation and Neuroprotection

Cannabinoids, both endogenous and exogenous, have been demonstrated to have anti-inflammatory, anti-oxidant, and neuroprotective effects in the CNS [309,310,311,312,313]. Glial cells, both astrocytes and microglia, are well-known to have potent immunomodulatory roles in the brain [314,315,316]. Activation of the glial CB1R and the CB2R has been demonstrated to promote an anti-inflammatory state by elevating anti-inflammatory cytokines and also lowering the levels of pro-inflammatory cytokines [317,318,319]. Activation of the astroglial CB1R has also been implicated in conferring protection to astrocytes against ceramide-induced elevation in free radicals and apoptosis [320,321]. However, CB1R antagonists were also reported to effectively attenuate various inducers of neurotoxicity, suggesting a far more complex mechanism by which CB1Rs confer neuroprotection [322,323]. Interestingly, both CB1Rs and CB2Rs are involved in abolishing lipopolysaccharide (LPS)-mediated pro inflammatory effects in astroglial cell culture [324], as well as in activated microglial cells [325,326,327,328]. The CB2R has also been shown to be upregulated in microglial cells when stimulated with inflammatory cytokines, and thereby has a significant role in the molecular underpinnings of pathological conditions characterized by neuroinflammatory states [85]. CB2R agonists have been shown to attenuate microglial activation when exposed to neuroinflammatory triggers [329], and also limit inflammatory responses at the BBB by reduction of tight junction protein expression in brain microvascular endothelial cells [330]. CB2R activation was also demonstrated to dampen inflammatory markers and reduce BBB permeability in post-traumatic brain injury [331,332,333]. The ability of the CB2R to modulate various aspects of neuroinflammation, involving both microglial activation and infiltration of peripheral immune cells, lends itself to be a valuable therapeutic target in neuroinflammatory diseases. For a comprehensive overview on the mechanisms of CB2R-mediated regulation of peripheral and central inflammatory states, please refer to the review by Cabral and Griffin-Thomas [334]. The role of CB2R in the regulation of peripheral immune cells is discussed in a later section. Schematic describing the role of astroglial, microglial, and lymphocytic cannabinoid receptors in the regulation of neuroinflammatory states and synaptic strengthening is illustrated in Figure 4.

#### 3.2.3. Implications of Dysregulated Endocannabinoid Tone in Neurological Disorders

Epilepsy: The fine tuning of excitatory and inhibitory synapses has important implications in epileptogenic activity. Evidence of THC demonstrating considerable anticonvulsant properties was studied prior to the identification of the endocannabinoid system [335,336]. After the discovery of the endocannabinoid system, the anticonvulsant properties of cannabinoids were shown to be associated with the brain CB1R activation [337]. CB1R activation has also been reported by multiple groups to mediate seizure termination and protection against excitotoxicity, gliosis, and brain damage [338,339,340,341,342,343]. In agreement with the above, SR141716A, a CB1R blocker, exhibited significant acute proconvulsant properties in various models of epilepsy [245,339,344]. Changes in CB1R expression in various forms of epilepsy have also been reported. An augmentation in CB1R expression in status epilepticus and febrile seizures [338,345], while a downregulation of CB1R protein and mRNA expression in excitatory axon terminals in hippocampal samples from patients with intractable temporal lobe epilepsy, has been reported [346]. Furthermore, in vivo PET human imaging further identified reorganization in CB1R availability in different brain regions following seizure onset [347]. CB1R expression was not only reported to be reorganized at various stages following status epilepticus [348,349], but its activation has been linked to region-specific differences in modulating synchrony [350].

However, the role of CB1R in regulating seizure activity is still unclear, and several factors such as age of onset may have a crucial role in determining the utility of employing an agonist versus an antagonist for treatment [351,352]. Opposing results from several studies have further complicated its role. CB1R-mediated protection against seizure activity was shown to be potentiated as well as neutralized by GABA-A modulators and agonists, suggesting a far more complex interaction with GABAergic system [351,352,353]. Contrasting with studies that reported the anticonvulsant properties of cannabinoids, both exogenous and endogenous cannabinoids have been reported to exhibit proconvulsant properties in various models of epilepsy [354,355,356]. While CB1R antagonism was shown to induce epileptogenic activity in a cell culture model of status epilepticus and in individuals with a history of epilepsy [357,358], CB1R blockade was shown to be efficacious in decreasing long term consequences of febrile seizures, suggesting potential benefits of preventing seizure-induced changes in endocannabinoid signaling [344]. Endocannabinoid signaling plasticity and the dual role of CB1R modulation in altering disease progression is discussed in detail elsewhere [359].

Apart from ambiguity over the CB1R’s role in the prevention of epileptic seizures, and the potential to induce psychological adverse effects, there is also some concern over the effectiveness of targeting CB1R solely as a therapeutic strategy [360,361]. By employing animal and cell culture models of epilepsy, several studies have demonstrated that CB2R is involved in conferring a protective phenotype to a variety of seizure types [362,363,364]. CB2R, which is primarily expressed on activated microglial cells, induced astroglial cell proliferation and survival in an astrocyte model of epilepsy [363]. Additionally, it was elevated in hippocampal neurons post pilocarpine-induced status epilepticus [365]. It could be that the CB2R could confer a neuroprotective phenotype by not only negating neuroinflammatory states, but also protecting neuronal and astroglial cells from toxic insults at various stages of epilepsy.

Parkinson’s disease: As mentioned earlier CB1R and other components of the endocannabinoid system are highly localized in the basal ganglia. Basal ganglia structures, such as the dorsal striatum, globus pallidus, and substantia nigra, play key roles in motor function. Failure to properly initiate or terminate synaptic activity in the basal ganglia neuronal circuitry has been linked to movement disorders such as Parkinson’s disease (PD) and Huntington’s disease (HD), which are characterized by hypo-and hyper--kinesia respectively. An augmented endocannabinoid tone, characterized by a significant increase in endocannabinoid and CB1R levels, were identified in the basal ganglia of animal models, and post-mortem brains of individuals with PD [366,367,368]. Higher levels of endocannabinoids were observed in the cerebral spinal fluid of untreated PD patients, which were normalized with dopaminergic treatment [369,370]. While promising results were observed in preclinical studies with SR141716A when administered with quinpirole [367], clinical studies with SR141716A failed to show any improvement in motor disability in PD patients [371]. CB1R agonists also showed mixed results. A pilot study demonstrated the efficacy of nabilone, a mixed CB1R/CB2R agonist, in alleviating symptoms in PD patients experiencing levodopa-induced dyskinesia [372]. On the other hand, a larger clinical trial demonstrated that cannabis was ineffective in ameliorating symptoms of levodopa-induced dyskinesia [373]. Biphasic changes in CB1Rs at early versus later stages of disease progression, in combination with brain structure-specific changes in CB1R expression, suggests a far more complex association of the endocannabinoid system with the progression of PD [368,374]. Interestingly, an open label observational study determined a significant improvement in motor symptoms in PD patients treated with medical marijuana [375]. This may suggest that medicinal marijuana, containing several cannabinoids, may offer greater therapeutic benefit in the treatment of PD, rather than a single CB1R agonist therapy. While “entourage effect” could be the most probable explanation, the involvement of CB1R-independent mechanisms in the progression of the disease should not be disregarded. HU-308, a CB2R agonist, has been previously shown to provide mild neuroprotection in rats with unilateral lesions of dopaminergic neurons [376]. Activation of CB2R in a MPTP mouse model of PD attenuated inflammatory markers and BBB damage, thereby protecting nigrostriatal neurons [377]. Targeting cannabinoid receptors other than CB1Rs can offer alternative routes to amelioration of symptoms in PD individuals [378].

Huntington’s disease: While PD is characterized by an augmented endocannabinoid tone, a significant reduction in CB1R expression, and also in endocannabinoid levels, has been reported in pre-clinical and clinical studies in HD [379,380,381,382,383,384]. In fact, a recent study reported that pre- frontal CB1R expression was reduced at very early stages of the disorder in individuals that carry the HD mutation, even prior to the onset of motor symptoms, suggesting that CB1R dysregulation may be an important factor in cognitive symptoms associated with HD [190]. Since cannabinoids can also counteract neuroinflammatory and pro-oxidant processes, apart from offering neuroprotection via negating excitotoxicity, employing cannabinoid-based therapeutics that target both CB1Rs and CB2Rs were viewed as viable strategies to limit neuronal damage often seen in HD. A pilot study examining the effect of nabilone in HD, reported an improvement in non-motor functions such as neuropsychiatric symptoms [385]. CB2R activation alone was also demonstrated to offer considerable neuroprotection, which was achieved by a reduction in the levels of pro-inflammatory markers in a rat model of HD [386]. However, a randomized, placebo-controlled, pilot trial examining the effects of Sativex, which comprises of THC and cannabidiol, did not show any improvement in motor, cognitive and behavioral functional scores when compared to the placebo [196]. More studies are needed to assess the efficacy of cannabinoid therapeutics in HD.

Alzheimer’s disease: Several groups have also investigated the utility of targeting the cannabinoid receptors in Alzheimer’s disease (AD) due to their role in neuroprotection. However, contradictory findings of changes in CB1R expression [387,388,389,390,391,392], and a lack of correlation between the CB1R and AD molecular markers or cognition [393], highlight a need to research non-CB1R components of the endocannabinoid system for AD. Brain CB2R have been reported to show significant correlation with AD markers, Aβ(42) levels, and plaque score [393]. CB2Rs are observed in activated microglia, the latter a prominent feature in inflammatory-related neurodegenerative diseases [393,394]. Apart from dampening of microglial activation [86], CB2R agonists have also been shown to stimulate Aβ clearance [395,396]. Given that CB2R agonists negate AD-related pathological outcomes via multiple mechanisms, and their relatively selective expression in activated microglia, they offer a potential alternative in the treatment of AD. This is however highly dependent on whether the preclinical outcomes of CB2R agonists hold true in a clinical setting. A novel CB2R agonist, NTRX-07, which has been shown to ameliorate neuroinflammatory states, is currently under Phase 1 trials for the treatment of AD [397].

Anxiety: Since CB1Rs are present in high abundance in prefrontal cortex, amygdala and hippocampus, the so called “emotional circuitry,” the effect of CB1R modulation has been investigated in reducing anxiety-like behavior. The CB1R has been demonstrated to play a role in controlling behavioral responses related to altered emotional states [398]. Cannabinoid agonists were reported to have a biphasic effect on modulating anxiety and stress, whereby low doses attenuate anxiety and higher doses bring about anxiogenic effects [399,400]. The molecular underpinnings of differential CB1R-mediated responses are diverse. These include contradictory functions of different brain regions in modulating anxiety [401], preferential activation of distinct CB1R populations on either GABAergic or glutamatergic neurons [402], and the potential activation of non-CB1R related mechanisms [403,404]. Additionally, environmental factors, such as stress attenuating GABA response to CB1R agonists, are also involved [405]. Apart from GABAergic and glutamatergic neurons, CB1Rs are also expressed in the raphe nuclei, whereby they can modulate functions of serotonergic neurons. Mice with serotonergic neurons deficient in CB1Rs, were shown to have heightened anxiety and reduced sociability [406]. Additionally, neuroinflammatory states have been also ascribed in the pathogenesis of anxiety and depression [407,408]. Since both the CB1R and the CB2R can reduce neuroinflammatory states, cannabinoid receptors provide multiple routes to induce anxiolytic effects. Evidence of CB2R polymorphisms in Japanese subjects with depression was reported, suggesting a link between the CB2R and behavior [409]. Additionally, in the same study, administration of antisense oligonucleotide targeting the CB2R mRNA expression in mice resulted in a reduction of anxiety-like behaviors [409]. By employing CB2R-over expressed mice and also spontaneously anxious mice, it was observed that the CB2R could alter anxiety-like behaviors via GABA receptors [410,411]. Deletion of the dopaminergic CB2R in the VTA of mice was also demonstrated to alter anxiety, depression, and psychomotor behavior [412]. Since CB1R antagonism by rimonabant has been associated with neurological adverse effects, alternate therapeutic avenues involving CB2R modulation may be a viable option.

Stroke: Cannabinoids have been demonstrated to exhibit neuroprotective functions against ischemic injuries via both the CB1R and the CB2R [413,414]. Cerebral ischemia-induced brain damage was observed to be more severe in CB1R knockout mice than the wild type mice, suggesting that the CB1R has neuroprotective roles post ischemia [414]. The protective role of CB1Rs in stroke is linked to not just the neuronal CB1R, but also involves the glial CB1R, whereby it has been shown to reduce glial cell reactivity and regulate LTD [415,416]. However, there is some ambiguity over CB1R’s neuroprotective role in cerebral ischemia [417]. In two separate studies, CB1R antagonists SR141716A and AM251 were shown to have beneficial effects in attenuating neuronal damage caused by cerebral ischemia [418,419]. Interestingly, CB2R agonists have also been shown to mitigate inflammatory states and promote neurogenesis in post stroke and post intracerebral hemorrhage [420,421]. The possibility of employing a dual CB1R antagonist/CB2R agonist strategy to attain effective coverage in attenuating the severity of ischemic injury, could be a viable therapeutic strategy based on the available pre-clinical data [422,423].

Neuropathic and inflammatory pain: Cannabinoids have been established to be potent analgesics [424,425]. Several studies have revealed that cannabinoids produce antinociception in diverse pathologies through both spinally and supraspinally located CB1Rs [426,427,428,429,430,431,432,433]. In pain models of inflammation, both CB1R and CB2R agonists showed considerable effectiveness in demonstrating analgesic effects [434,435]. Owing to the ability of CB2Rs to suppress microglial activation and neuroinflammatory states, CB2R agonists were also investigated as a potential therapeutic strategy to treat neuropathic pain. CB2R knockout mice showed enhanced interferon (IFN)-γ production by glial cells [436]. Consistent with this, the CB2R exhibited potent hyperanalgesic effects through both central and peripheral mechanisms [437,438,439,440,441]. Interestingly, CB2Rs were reported to be upregulated within the spinal cord in a neuropathic, but not an inflammatory, pain model [301]. However, CB2R agonists were reported to reduce thermal hyperalgesia associated with inflammation [442,443]. A recently performed meta-analysis of randomized controlled trials revealed cannabinoids produced a small, yet significant, improvement in pain reduction in individuals with neuropathic and non-neuropathic pain [444].

### 3.3. Metabolic

#### 3.3.1. Hypophagia

CB1R activation has been well-established to promote a feeding response. Reduced food intake was reported in rats administered CB1R antagonists and also in CB1R knockout mice, which suggests that tonically activated CB1R has a critical role in modulating food intake [445,446]. CB1Rs exert this effect through a multi-level regulation of appetite [447]. The primary level of regulation is at the hypothalamic nuclei, where it modulates the release of orexigenic and anorexigenic neuropeptides [448,449]. The CB1R has been shown to be essential for both leptin’s and ghrelin’s effects on food intake [446,450]. Apart from the hypothalamic nuclei, the CB1R is also reported to regulate food intake and energy homeostasis at the level of the brainstem nuclei [451]. Lastly in the mesolimibic reward system, whereby CB1R agonism promotes reward effects by elevating dopamine release [452]. However, the role of CB1R-mediated hypophagic action is far more complex, since factors such as the current prandial state and fluctuations in endocannabinoid levels and/or CB1R-mediated signaling, also impact its effect on appetite regulation [453].

#### 3.3.2. Peripheral Energy Regulation

While the central CB1R has well-established roles in the regulation of appetite by rewiring neuronal circuitry to facilitate rewarding behaviors, peripheral CB1Rs play a key role in the homeostatic control of metabolites. Importantly, the CB1R has been implicated in the development of obesity [454]. CB1R antagonism improved insulin signaling in pancreatic cells, improved β-cell proliferation and improved glucose responsiveness by increasing glucokinase and glucose transporter 2 (GLUT-2) expression in pancreatic β cells [455,456]. Ibipinabant, a selective CB1R antagonist, exhibited significant antidiabetic effects, such as attenuation of β-cell loss in male Zucker diabetic fatty rats which was independent of its effects on weight loss [457]. Additionally, β-cell-specific CB1R ablation has been shown to reduce a heightened inflammatory state in the pancreatic cells in response to a high fat/sugar diet [458]. Although multiple lines of evidence have shown the importance of the pancreatic CB1R in preventing β-cell loss and improving function, activation of the CB1R expressed on infiltrating macrophages was shown to be associated with pancreatic β-cell failure [459]. For a detailed mechanistic overview on the role of pancreatic CB1R in diabetes, please refer to the review by Jourdan et al. [460].

Apart from the pancreas, functional CB1Rs were reported in various insulin-sensitive tissues such as liver, adipose, and skeletal muscle. Similar to the pancreatic CB1R, hepatic CB1R deletion was associated with a positive outcome; whereby a reduction in steatosis and hepatocellular damage, and an improvement in glucose tolerance and insulin resistance was observed [90]. In the case of adipocytes, CB1R activation was associated with altered metabolic profiles such as augmented lipogenesis and surges in plasma triglycerides and cholesterol in vivo [461,462]. CB1R activation was also reported to stimulate peroxisome proliferator-activated receptor gamma (PPAR-γ) and decreased adiponectin in adipocytes [463], while blockade was shown to elevate Slc2a4/GLUT4 expression in adipocytes [464]. Peripheral CB1R blockade also improved insulin resistance in adipose tissue in diet-induced obese mice by mitigating an inflammatory response [465]. Additionally, CB1R blockade was shown to improve mitochondrial biogenesis and elevated nNOS expression in cultured white adipocytes, while agonism impaired mitochondrial biogenesis and decreased AMPK phosphorylation [466,467]. CB1R antagonism was also reported to recalibrate the abdominal muscle glycolytic and TCA cycle pathways in response to a hypercaloric diet [468]. In the case of skeletal muscles, CB1R antagonism was shown to upregulate GLUT-4 and improve intermittent hypoxia-induced insulin resistance [469], while CB1R agonism was shown to interfere with insulin signaling pathways [470,471]. Additionally, CB1R antagonism was shown to affect mRNA expression of genes involved in skeletal muscle oxidation in myotubules in obese individuals [472]. This suggests that the CB1R is well placed to have important modulatory effects on peripheral energy homeostasis.

#### 3.3.3. Implications of a Dysregulated Endocannabinoid Tone in Metabolic Disorders

Metabolic disorders and associated comorbidities: Since both central and peripheral CB1Rs are involved in the regulation of appetite and metabolic processes, CB1R antagonism was investigated as a potential therapeutic opportunity for obesity and other metabolic disorders. While multiple trials confirmed the efficacy of rimonabant in weight loss and improvement of metabolic risk factors, an elevated risk of developing neuropsychiatric adverse effects has led to the discontinuation of this drug [473,474,475,476]. The association of neurological complications with rimonabant treatment, is a major impediment for the use of CB1R antagonists as a therapeutic strategy [205,477]. Since the peripherally selective CB1 antagonist TM38837 was reported to elicit minimal CNS effects in humans [478], peripheral CB1R blockade may well be a viable therapeutic option for targeting metabolic disorders. In addition to promoting weight loss [479,480,481,482,483], peripherally selective CB1R antagonist/inverse agonist also demonstrated improvements in metabolic profiles [465,484,485] and liver function [486] in preclinical studies, thereby further highlighting the therapeutic utility of targeting the peripheral CB1R in metabolic syndromes. Interestingly, a dual therapy comprising of a peripheral CB1R antagonist with a CB2R agonist, was shown to be a highly effective strategy in the prevention of diabetes-related comorbidities in streptozocin-induced diabetic rats [487]. By combining CB1R antagonists with drugs that have complimentary roles, peripheral cannabinoid therapeutics could be an attractive approach in the treatment of metabolic disorders. It should be noted that evidence of beneficial effects of endocannabinoid activation in metabolic syndrome has also been reported. Analysis of NHANES data revealed a lower incidence of metabolic syndrome in individuals with marijuana use [488]. Since marijuana comprises of several types of phytocannabinoids, mechanisms other than CB1R activation cannot be discounted in that study.

Cachexia: Cachexia, one of the more debilitating consequence of wasting syndrome, is characterized by a loss of fat and muscle tissues. In addition to a decrease in food intake, metabolic abnormalities such as a shift from anabolic to catabolic processes are believed to result in tissue degeneration [489,490]. As cannabinoids have a profound effect on appetite and metabolism, they have been widely employed to stimulate appetite in pathological conditions such as cachexia. In this regard, THC and nabilone have been approved for the treatment of cachexia [491]. The CB1R is believed to have a critical role in cachexia, because of its ability to stimulate appetite, induce lipogenesis, fat storage, and induce weight gain [492]. Although anecdotal evidence for the utility of cannabinoids in cachexia is abundant, direct evidence of cannabinoid related improvement of quality of life in individuals with cachexia surprisingly is still not definitive [493,494].

### 3.4. Cardiovascular

#### 3.4.1. Triple/Triphasic Response of Cannabinoids

Since the CB1R is expressed in the heart, blood vessels, and the cardiovascular centers of the brainstem and hypothalamus, cannabinoids can exert an influence on blood pressure regulation through multiple mechanisms [474,495]. Intravenous administration of cannabinoids, both exogenous and endogenous, triggers an immediate fall in blood pressure which is vagally mediated (Phase I), followed by a brief pressor effect (Phase II), and this gives way to a prolonged hypotensive effect (Phase III) [496,497]. While TRPV-1 is implicated in mediating cannabinoids-induced reflex bradycardia (phase I), the prolonged hypotensive effect (phase III) is mediated via peripherally located CB1R, although central effects cannot be disregarded [498,499,500,501]. For a comprehensive review of cannabinoid-mediated triphasic effect, please refer to the review by Malinowska et al. [495].

#### 3.4.2. Central Mechanisms of Blood Pressure Regulation

The cardioregulatory centers in the medulla are implicated in the short and long-term regulation of blood pressure. The peripherally located baroreceptors senses increases in blood pressure and relay the information to the critical command center, NTS. This in turn activates the parasympathetic tone, and inhibits the pressor center, RVLM. Thus, a decrease in baroreflex sensitivity, often results in a reduced buffering capacity to a spike in blood pressure. Over time this results in baroreflex resetting; often an essential pathophysiological alteration in essential hypertension [502]. For an overview of general central mechanisms that regulate blood pressure, please refer to detailed reviews on this subject by Fisher [503,504] and Haspula and Clark [505]. Central administration of cannabinoids has been shown to elicit either sympathoinhibition or sympathoexcitation based on the site of microinjection. For instance, microinjection of cannabinoids into the NTS of rats can enhance baroreflex responses and triggers sympathoinhibition [506,507,508,509,510], partially by altering GABA release and baroreflex sensitivity [506,510]. However, microinjection of cannabinoids into the midbrain periaqueductal gray and also the RVLM of rats resulted in increased sympathetic activity, and increased blood pressure [511,512,513]. The intracellular signaling kinases PI3K and ERK1/2, as well as nNOS have been implicated as potential mechanisms in cannabinoid-induced pressor responses in the RVLM [514,515]. Additionally, enhanced orexin receptor 1 signaling has been described as a priming event for the central CB1R-mediated pressor response [516].

#### 3.4.3. Peripheral Mechanisms of Blood Pressure Regulation

Although the CB1R can modulate heart rate via central mechanisms, cardiac CB1R activation was shown to elicit negative chronotropic and ionotropic effects, that were potentially independent of the CNS [95,517,518]. Mechanistic studies revealed that the PLC-cGMP and NOS pathways were crucial signaling pathways for mediating the contractile responses of CB1R [519]. CB1R activation was also shown to prevent cardiac remodeling in cultured rat cardiomyocytes by the suppression of phosphorylation of MAPKs and EGFR [520]. Interestingly, this CB1R-dependent cardiodepressant effect is augmented only in individuals with pathological conditions, and not in healthy individuals [96,518,521]. This was evident in CB1R knockout mice, where blood pressure and heart rate remained normal suggesting a lack of tonic control over these cardiovascular indices [522]. But in a mouse model of congestive heart failure and also acute heart failure, CB1R deficiency was associated with an increase in mortality rate [520,523]. Regarding the cardiodepressant mechanism, a possible involvement of peripheral TRPV1 channels cannot be disregarded [524]. Apart from direct cardiodepressant effects, several studies have reported vasodilatory effects of cannabinoids in the aorta and coronary arteries through both CB1R-dependent and independent mechanisms [525,526,527,528,529,530]. Anandamide was demonstrated to activate two distinct signaling streams via the CB1R and the endothelial atypical receptor or GPR55 based on the activation and clustering of integrins in human endothelial cells [531]. This involved CB1R-mediated translocation of NFκB via spleen tyrosine kinase, and alternatively GPR55-mediated calcium mobilization via PI3K-PLC pathway [531]. Interestingly, anandamide was also shown to decrease endothelin-1 production and increase nitric oxide levels via a non-CB1R mechanism in human endothelial cells [532]. For a comprehensive review on the effects of cannabinoids through non-CB receptor mechanisms in the vasculature, please refer to the review by Bondarenko [533].

#### 3.4.4. Implications of a Dysregulated Endocannabinoid Tone in Cardiovascular Disorders

Hypertension: The use of cannabinoids as potential anti-hypertensive medications has mixed results in a clinical setting. Studies in the 1970s reported hypotensive effects of chronic cannabinoid use in humans. Prolonged use, either by marijuana inhalation, or by THC consumption, resulted in a significant fall in heart rate and blood pressure [534,535]. THC was also demonstrated to have a potent hypotensive effect in hypertensive individuals when compared to normotensive individuals [536]. Multiple studies have also reported evidence of central sympathoinhibition in response to cannabinoids [88,537]. However more recently, mixed results were reported on the modulation of cannabinoid receptor activity. While the cross-sectional analysis of NHANES identified a lower prevalence of metabolic syndrome in marijuana users, a higher occurrence of elevated systolic blood pressure was reported [488]. Interestingly, no differences in blood pressure were noted when rimonabant-treated groups were compared with the control groups over a period of 2 years [473].

By employing SHRs and other rat models of hypertension, several studies in the late 1990s were able to elucidate the mechanisms by which cannabinoids alter cardiovascular function in hypertension. Systemic administration of cannabinoids was observed to have a mild hypotensive effect on blood pressure regulation in normotensive rat models. However, in hypertensive rat models, a greater reduction in blood pressure when compared to the normotensive models in response to cannabinoids was observed [96]. This enhancement of basal endocannabinoid tone resulted in improved cardiovascular parameters, such as heart rate and vascular resistance, in SHRs and other models of hypertension [96]. Interestingly, while the myocardial and endothelial CB1Rs were elevated in SHRs [96], a reduced density of CB1Rs was reported by multiple groups in SHR brains [254,538,539], and a dampened endocannabinoid tone was reported in the CNS [538]. Centrally administered cannabinoids had a marked sympathoinhibitory effect in normotensive rat models such as WKY and Sprague Dawley rats, but not in SHRs [538]. Results from the aforementioned pre-clinical studies suggest a differential alteration in CB1R functionality in hypertensive conditions at the level of the peripheral organs, when compared to the CNS. In this regard, endocannabinoid hyperactivity in the periphery could be an adaptive or a compensatory mechanism in response to an elevation in blood pressure triggered by sympathetic augmentation. Hypofunctional endocannabinoid system in the NTS resulting in an elevated sympathetic activity in SHRs could in fact be a crucial mechanism early on in the development of hypertension. This is a plausible theory since sympathetic hyperactivity is theorized to be a critical cardioregulatory modification that occurs at earlier stages of hypertension [505,540]. Although several studies have also reported a blood pressure-lowering effect on cannabinoid receptor activation, sympathoexcitatory effects of endocannabinoid system activation have also been observed [511,541]. In this case, microinjection of the CB1R antagonist into the NTS was shown to improve baroreflex sensitivity and reduce systolic blood pressure in (mRen2)27 rats [542,543]. The apparent discrepancy in results in clinical and pre-clinical settings suggests a more complex relationship between CB1Rs and blood pressure regulation in hypertension. Some of the reasons could be due to the pleiotropic effects of cannabinoids, and differences in experimental design such as the use of unanesthetized versus anaesthetized animals, and the use of monogenic or polygenic rat models of hypertension.

Cardiac, endothelial, and vascular dysfunction: Since CB1R activation exerts cardiodepressant and protective roles, cardiac CB1Rs could be potentially modulated for improving cardiac function. Although CB1R agonism has been shown to have beneficial effects in animal models of hypertension and heart failure [96,520,523], CB1R antagonism was shown to improve cardiac function after experimental myocardial infarction and metabolic syndrome [544]. Additionally, both pharmacological and genetic inhibition of CB1R improved cardiac function and resulted in an attenuation of oxidative stress, inflammation, and fibrosis in a rat model of diabetes [545]. CB1R inhibition in apolipoprotein E-deficient mice on a cholesterol-rich diet resulted in improvement of endothelial parameters and function, such as a decreased AT1R expression and ROS production [546]. This is in accordance with previous studies on the beneficial effects of rimonabant on improvement of cardiovascular and metabolic parameters in metabolic syndrome. However, G_q/11_-mediated vascular endocannabinoid synthesis, and the resultant CB1R activation in vascular smooth muscle cells was shown to attenuate Ang II-mediated vasoconstriction [89,547]. It could be that therapeutically activating or inhibiting cardiac and vascular CB1R is dependent on whether cardiac dysfunction is present with comorbidities such as hypertension or diabetes. However, the use of synthetic cannabinoids for cardiovascular diseases needs to be researched further, since they have been associated with various cardiac events in adults [548]. Some of the most important biological functions of the cannabinoid receptors discussed in this review thus far, are shown as a schematic in Figure 5.

### 3.5. Modulation of Immune System

#### 3.5.1. Immune Cell Development and Differentiation

The CB2R, which is present primarily on immune cells, plays an integral role in the regulation of both humoral and cell-mediated immunity. Cannabinoids are well-known to possess potent immunosuppressive properties [319,549]. Cannabinoids via CB2Rs have been demonstrated to suppress T- and B-cell proliferation and function, and induce apoptosis of splenocytes and thymocytes [550,551,552]. Deficiency of CB2Rs has been demonstrated to cause an augmentation of inflammatory state and chemotactic functioning [553]. However, CB2R deficiency was also shown to reduce T- and B-cell population [554], and also resulted in an impaired B-cell retention and function [555]. In support of this, lower concentrations of synthetic cannabinoids were shown to also enhance proliferation and maturation of B lymphocytes potentially via the CB2R [556], suggesting a more complex role for the CB2R in immunomodulation [557]. Varying levels of CB2Rs were also reported among the various immune cell subpopulations [100,558,559]. Additionally, CB2R expression is regulated by the type of stimulus and the activation status of immune cells [559]. Stimulation of CB2R in activated macrophages was shown to attenuate LPS/IFNγ-induced interleukin (IL)-12p40 release, and elevated IL-10 release [560].

#### 3.5.2. Implications of a Dysregulated Endocannabinoid Tone in Immunological Disorders

Rheumatoid arthritis: Rheumatoid arthritis (RA) is an inflammatory disease that is characterized by an augmented infiltration of immune cells in the synovial cavity. In addition to regulating various aspects of T- and B-cell functions, the CB2R is also found in various other cell types involved in RA pathology, such as synovial tissue [561], osteoclasts [562], and chondrocytes [563]. Interestingly, the CB2R was upregulated in fibroblast-like synoviocytes in response to pro-inflammatory triggers such as IL1-β and TNFα [564]. Furthermore, a CB2R agonist negated IL1-β and TNFα-induced elevation in proinflammatory cytokines and matrix metalloproteases in RA [564,565]. Additionally, activation of the CB2R was also shown to negate several proinflammatory mediators and inhibit RA-induced bone damage in collagen-induced activation in mice [566]. While the anti-inflammatory effects observed with CB2R agonists were most likely mediated via the CB2R, involvement of other mechanisms or receptors, such as the glucocorticoid receptor, cannot be ruled out [567].

Autism spectrum disorder-mediated immune impairment: Clinical and pre-clinical studies have routinely reported immune dysfunction in autistic individuals [568,569]. While the CB1R was unchanged in peripheral blood mononuclear cells in children with autism, both CB2R protein and mRNA were significantly upregulated [570]. In a follow-up study, Gc protein-derived macrophage-activating factor treatment, which was shown to have beneficial effects in the treatment of autism, was able to reduce CB2R expression in bone-marrow-derived macrophage cells from autistic individuals, and reduce macrophage activation [571]. Although these results are promising, further studies are required to ascertain whether the CB2R can be therapeutically targeted for normalizing immune functions in autistic individuals.

Multiple sclerosis: Multiple sclerosis (MS) is an autoimmune disorder characterized by demyelination of nerve fibers in the brain and spinal cord. Evidence of a role for cannabinoid receptors in the pathogenesis of MS comes from multiple in vitro studies that employed an experimental autoimmune encephalomyelitis (EAE) model of MS. Both CB1Rs and CB2Rs were altered in immune cells isolated from primary-progressive and relapsing–remitting MS patients [195,572]. Additionally, both CB1Rs and CB2Rs were also involved in neuroprotection against excitotoxicity in a chronic model of MS [573]. Postmortem brain samples taken from MS individuals showed high CB1R and CB2R immunoreactivity in non-neuronal cells, such as oligodendrocytes, microglia, and macrophages, from active plaques [574]. Since glial inflammation and infiltration of inflammatory cells is a key factor of MS, targeting only CB2R to attenuate neuroinflammatory states would be devoid of CB1R-related psychological adverse effects [575]. Interestingly the CB2R, but not the CB1R, was reported to be upregulated in activated microglial cells and peripheral macrophages in EAE model of MS [85], and was demonstrated to have critical roles in inhibiting leukocyte/endothelial interactions and attenuating infiltration of inflammatory cells [576]. Additionally, CB2R-selective agonists were shown to exert potent anti-inflammatory and anti-proliferative effects on immune cells isolated from MS patients [577,578]. WIN-55,212-2 was reported to reduce inflammatory infiltrates in spinal cord and also induced apoptosis of encephalitogenic T-cell populations, partially via the CB2R in EAE model [579]. However, CB1R-related therapies cannot be disregarded since the CB1R has also been reported to treat various symptoms of MS, such as spasticity [580,581]. Interestingly, CB1Rs expressed on neuronal cells, but not on T cells, were also shown to have a critical role in cannabinoid-mediated attenuation of EAE pathology [582].

Atherosclerosis: Since atherosclerosis is a chronic inflammatory disease characterized by macrophage infiltration, the potential of targeting cannabinoid receptors was investigated using animal models of atherosclerosis. Both phytocannabinoids and synthetic cannabinoids have been demonstrated to exert potent anti-inflammatory and anti-atherosclerotic effects via the CB2R in preclinical studies [583,584]. CB2R knockout in mice with hyperlipidemia-induced atherogenesis displayed greater macrophage and matrix metalloproteinase content when compared to their controls [585]. Apart from cannabinoid treatments altering metabolic and lipid levels, the CB2R agonism was also shown to have potent anti-chemotactic effects via alteration of chemokine receptor and adhesion molecule expression in monocytes [586].

Pain: Exacerbated inflammatory responses are known to be associated with both neurogenic and non-neurogenic pain states [587,588]. The role of central CB1R, and both central and peripheral CB2R inducing anti-nociceptive actions in neuropathic and inflammatory models of pain has been described in earlier (Section 3.2.2 and Section 3.2.3). Additionally, recently conducted retrospective and meta-analysis revealed the utility of cannabinoids in the treatment of postoperative pain and other forms of acute pain states [589,590]. Interestingly, the peripheral CB1R has also been implicated in inducing anti-nociception and resolving associated inflammatory states in various animal models of pain [591,592,593,594]. Since THC has been shown to have tremendous anti-inflammatory potential, which includes its actions on prostaglandin synthesis [595,596], designing peripherally-restricted cannabinoid therapies for the treatment of pain can be a potential alternative to opioid-based therapies [597].

## 4. Therapeutic Strategies

Strategies to alter endocannabinoid tone in pathological conditions, with minimal-to-no adverse effect profiles, are highly desirable. This is especially true, since targeting central CB1R has been associated with severe neurological adverse effects. This has to be achieved by either employing a direct strategy, which involves either activating or inactivating cannabinoid receptors using mono- or combination-drug therapies, orindirectly, by inhibiting degradative enzymes of endocannabinoids resulting in enhanced cannabinoid receptor activation [359,598,599,600].

### 4.1. Multi-Drug Strategy

Factors that have been implicated in the pathogenesis of cardiovascular, metabolic, autoimmune, and neurological disorders, such as Ang II and inflammatory cytokines, have all been implicated to intrinsically modulate the endocannabinoid tone by altering cannabinoid receptor expression or activity, through receptor crosstalk mechanisms [89,123,139,153,539]. As a result, employing low dose partial CB1R agonists/ neutral antagonists as adjuvant therapies to existing drug therapies could potentially limit the neuropsychiatric adverse effects that are often linked to high dose of CB1R monotherapies. Combination therapies were suggested by numerous studies based on strong preclinical data [542,601]. In addition to dual drug therapy, the ability of the CB1R to dimerize with other receptors can also be leveraged to design novel therapies. The CB1R has a high tendency to crosstalk with several different GPCRs, including but not limited to the AT1R and the opioid receptors [141,145,602,603,604]. Since receptor heteromers are selectively expressed in only distinct pathological conditions [605,606], targeting heteromeric complexes has been discussed as a viable therapeutic strategy [607]. Additionally, targeting CB1R heteromeric complexes using divalent ligands would result in selective targeting of tissues or brain regions expressing dimers, without altering CB1R-based signaling in other regions. The design and synthesis of divalent ligands for CB1R-orexin receptor and CB1R-D2R heteromer have already been reported [608,609].

### 4.2. Peripheral CB1R Antagonists

Another strategy is to employ peripherally restrictive CB1R antagonists that would considerably enhance the safety profile of the drugs. Second and third generation CB1R antagonists, that have enhanced peripheral selectiveness, have already demonstrated significant therapeutic potential in preclinical studies [610]. Although peripheral CB1R modulators may possess greater benefit-to-risk ratio than those that cross the BBB, the former may be lacking in effectiveness for the treatment of neurological disorders characterized by a dysregulated endocannabinoid tone.

### 4.3. Allosteric Modulators

Nonselective activation/deactivation by orthosteric ligands are known to elicit both therapeutic and unwanted effects of CB1R. The utility of allosteric modulators of the CB1R or neutral CB1R antagonists, as an alternative for CB1R orthosteric ligands and inverse agonists respectively, could be useful to circumvent the neuropsychiatric adverse effects associated with CB1R-based therapies [611,612]. Studies on the negative CB1R allosteric modulators, Org27569 and PSNCBAM-1, have already been reported [118,158,613]. The design of the newer generation negative allosteric modulator, such as GAT100, may open up unprecedented avenues for CB1R drug discovery since it demonstrated enhanced potency with negligible inverse agonism properties in functional cell assays [614].

### 4.4. CB2R Modulators

Considering that the CB2R does not participate in normal brain and peripheral tissue function under healthy conditions, and it is only present in pathological states, targeting the CB2R in neurological disorders characterized by neuroinflammation could be a viable therapeutic strategy to leverage neuroprotective effects of the endocannabinoid system [615]. Additionally, the CB2R displays a reduced adverse effect profile, as CB1R-mediated tolerance and physical dependence with repeated doses is circumvented [616]. In addition to employing multiple drugs, more recently, “dual-mechanism” drugs, such as TV-6–41, which possess both CB2R agonism and CB1R neutral antagonist properties have displayed a reduced adverse effects profile when compared to rimonabant in preclinical studies [617]. Such drugs may be valuable in increasing the therapeutic effectiveness of cannabinoid receptors while significantly attenuating CB1R-mediated unwanted effects. While CB2R-based therapies have shown promise in animal studies, their inability to translate effectiveness from preclinical to clinical studies remains a major hurdle [616,618,619]. Although several reasons have been previously highlighted, one important factor could be the lack of specific commercially available antibodies for the CB2R which results in an over-reliance on gene expression data to develop therapeutics. Since a lack of significantly high correlations between gene and protein expression exists [620], inferences drawn solely from CB2R gene expression studies in preclinical studies need to be confirmed with other additional functional assays.

### 4.5. Inhibitors of Degradative Enzymes

Because of the adverse effects on the CNS, the adverse effect profile of exogenous cannabinoids is a limiting factor. To circumvent this issue of dose-dependent neurological adverse effects, inhibitors of endocannabinoid degradative enzymes, monoacylglycerol lipase (MAGL) and FAAH, were employed to augment the endocannabinoid tone. ABD-1970, a potent MAGL inhibitor, was demonstrated to exhibit significant antinociceptive effects without the behavioral effects that would be observed with exogenous cannabinoids [621]. PF-04457845, also a FAAH inhibitor, was well-tolerated with limited efficacy in osteoarthritis [622,623]. However, phase-1 clinical trials employing BIA 10-2474 (another FAAH inhibitor) resulted in lethal adverse effects with the use of the highest dose of the drug [624]. This suggests that further research on the safety and efficacy of optimal dosing of indirect cannabinoid therapies is needed.

## 5. Conclusions

Since the discovery of the endocannabinoid system, there has been a major surge in publications on medical cannabis, cannabinoid-based therapeutics, and cannabinoid receptor pharmacology [625,626]. These studies have aided greatly in deciphering the endocannabinoid system in the molecular and the physiological realms [18,627]. Recent studies have brought to light the versatility and malleability of context-dependent CB1R signaling [628,629,630]. Identifying factors and conditions that could regulate CB1R expression and *CNR1* gene expression could therefore help circumvent the need for direct CB1R agonists/antagonists. Since overcoming the neuropsychiatric adverse effects of CB1R-directed agonists/antagonists remains a challenge, CB2R-based therapeutics could be the path forward. Not only do both the CB1R and the CB2R have complimentary roles in various pathological conditions, but they also exhibit contrasting/unique roles in several disease states. The utility of CB2R-based therapeutics could also be greatly enhanced by combining them with pre-existing treatment options instead of a stand-alone therapeutic strategy [631]. Additionally, indirect modulators of endocannabinoid tone may be another promising avenue. Regardless of the promising therapeutic efficacy demonstrated by the aforementioned strategies, safety profiles of newer cannabinoid therapeutics remain an ever-present concern [624]. A multidisciplinary approach could also be utilized in personalizing cannabinoid therapeutics based on genetic variables associated with basal endocannabinoid tone [632,633]. As we embark into the fourth decade of endocannabinoid research, the future of cannabinoid-based therapeutics still holds a lot of promise

## Figures and Tables

**Figure 1 ijms-21-07693-f001:**
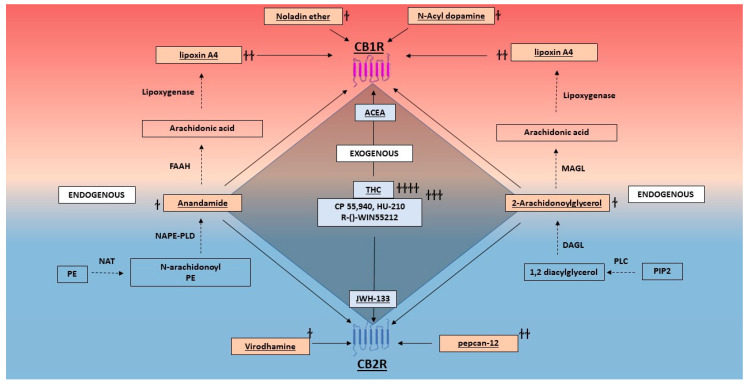
Overview of endogenous and exogenous cannabinoids that activate/modulate CB1R and CB2R. An overview of CB1R activators and modulators are shown in this figure. This includes endogenous (**ɫ**,**ɫɫ**), and exogenous (**ɫɫɫ**,**ɫɫɫɫ**) ligands. Examples of endogenous ligands listed here are the endocannabinoids (2-AG, anandamide, N-acyl dopamine, noladin ether and virodhamine (**ɫ**)); endogenous positive allosteric modulators (lipoxin A4 and pepcan-12 (**ɫɫ**)); synthetic exogenous cannabinoids for CB1R (ACEA (**ɫɫɫ**)), CB2R (JWH133 (**ɫɫɫ**)) and both (CP 55,940, HU-210 and R-()-WIN55212 (**ɫɫɫ**)); and phytocannabinoids (Δ9-tetrahydrocannabinol (**ɫɫɫɫ**)). (PE: phosphatidylethanolamine; NAT: N-acetyltransferase; NAPE-PLD: N-acyl phosphatidylethanolamine phospholipase D; FAAH: fatty acid amide hydrolase; PIP2: phosphatidylinositol 4,5-bisphosphate; PLC: phospholipase C; DAGL: diacylglycerol lipase; MAGL: monoacylglycerol lipase; ACEA: arachidonyl-2′-chloroethylamide; THC: Δ^9^- tetrahydrocannabinol).

**Figure 2 ijms-21-07693-f002:**
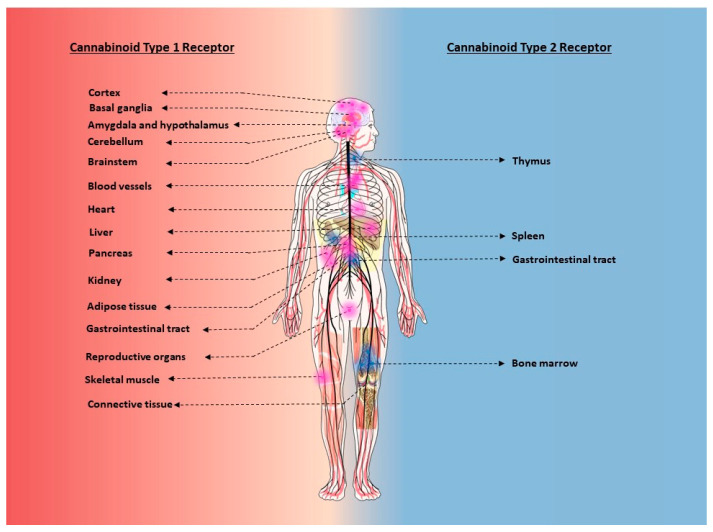
Tissue distribution of CB1R and CB2R under normal physiological conditions. The figure shows tissue expression of CB1R and CB2R under healthy conditions. Although CB1R protein or transcripts have been identified in several non-neural tissues, they are predominantly localized in the central nervous system (CNS). On the other hand, the CB2R is predominantly found in tissues involved in immune regulation. It is sparsely distributed in the CNS under normal physiological conditions.

**Figure 3 ijms-21-07693-f003:**
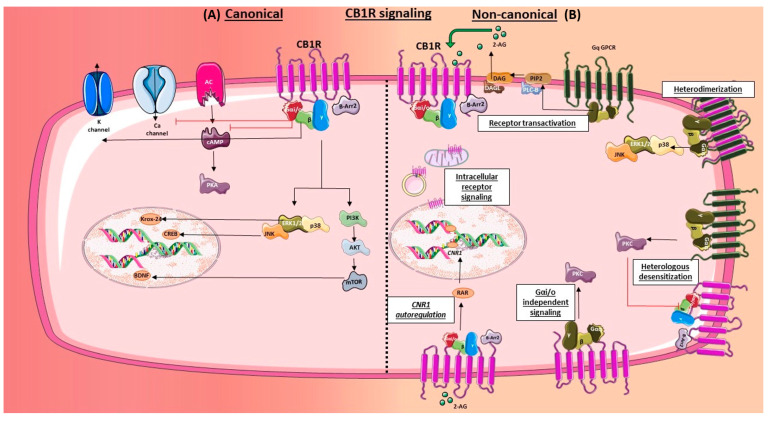
Canonical and non-canonical signaling of CB1R. (**A**) Canonically, CB1R activation results in coupling of pertussis toxin (PTX)-sensitive G-protein (Gα_i/o_), and the activation of GIRK and inhibition of calcium channels. CB1Rs also activate MAPKs, such as ERK1/2 and JNK, which result in the subsequent induction of Krox-24 and CREB respectively. In addition, CB1R stimulation also leads to the downstream activation of PI3K/AKT/mTOR pathway, which further results in the transcription of BDNF. (**B**) The CB1R is also involved in non-canonical signaling. Gα_q/11_ GPCR-mediated-mobilization of endocannabinoids, and subsequent activation of the CB1R in an autocrine or paracrine fashion (Receptor transactivation). The CB1R has also been shown to dimerize with other GPCRs resulting in a change in CB1R-mediated signaling, such as changes in MAPK activation patterns (Heterodimerization). Additionally, GPCRs that activate PKC, such as Gα_q/11_ GPCRs, have also been shown to phosphorylate CB1R and potentially dampen its activity in certain cell types (heterologous desensitization). Certain cannabinoids have been shown to couple to Gα_q/11_ and Gα(s) proteins, and effectively activate calcium channels (Gαi/o independent signaling). In addition to being membrane-bound, functional CB1R has also been reported to localize intracellularly, such as the nucleus and the mitochondria, where they are capable of signaling (intracellular signaling). Finally, cannabinoids have been shown to trigger an induction of *CNR1* by the activation of CB1R in various cell types (autoinduction). (AC: adenylyl cyclase; cAMP: cyclic adenosine monophosphate; PKA: protein kinase A; β-Arr2: Beta-arrestin-2; ERK1/2: extracellular signal-regulated kinases; JNK: c-Jun N-terminal kinases; PI3K: phosphoinositide 3-kinases; AKT: protein kinase B; mTOR: mammalian target of rapamycin; PIP2: phosphatidylinositol 4,5-bisphosphate; PLC-β: phospholipase C beta; DAG: diacylglycerol; DAGL: diacylglycerol lipase; 2-AG: 2-arachidonoylglycerol; RAR: retinoic acid receptor; *CNR1*: cannabinoid receptor 1 gene).

**Figure 4 ijms-21-07693-f004:**
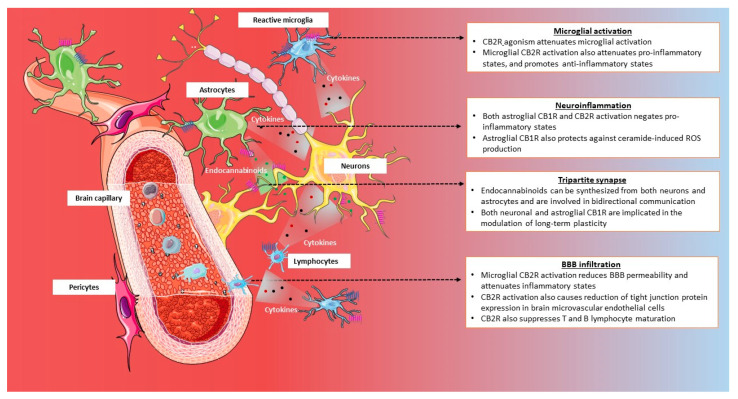
Regulation of synaptic activity and neuroinflammatory states by cannabinoid receptors. CB2R located on both microglial and astroglial cells are involved in the attenuation of inflammatory states. They do so by inhibition of microglial activation, reduced secretion of pro-inflammatory cytokines, and increased secretion of anti-inflammatory cytokines, and limiting the infiltration of peripheral immune cells. The CB2R is also expressed in the brain microvascular endothelial cells whereby they regulate expression of tight junctions, and further limit chemotaxis and transmigration of peripheral immune cells into the CNS. The CB2R also limits T- and B-cell proliferation and immunomodulation. Astroglial CB1R also promotes an anti-inflammatory state while simultaneously lowering levels of pro-inflammatory cytokines. Additionally, both neuronal and astroglial cells secrete endocannabinoids which are involved in modulation of synaptic strengthening by LTP and LTD. Additionally, astroglial CB1R activation has also been demonstrated to protect against excitotoxic neuronal damage and some forms of neurotoxic damage.

**Figure 5 ijms-21-07693-f005:**
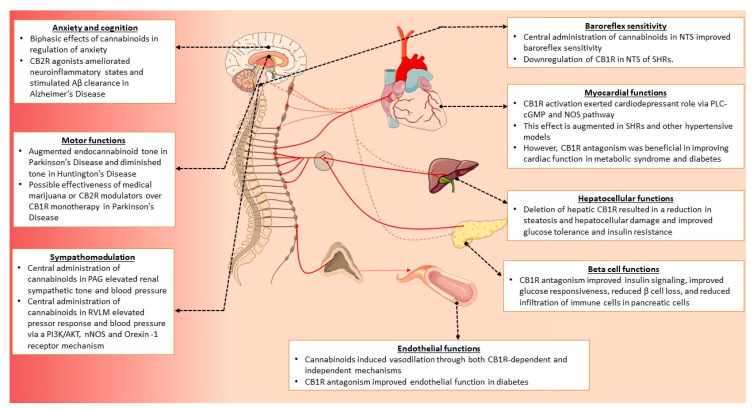
Functional overview of cannabinoid receptors in the CNS and peripheral tissues. CB1R is densely expressed in regions of the brain involved in the regulation of anxiety and cognition such as prefrontal cortex, amygdala, and hippocampus. Activation of the CB1R has been shown to have both anxiogenic and anxiolytic effects. Endocannabinoid tone is also altered in basal ganglia disorders such as Parkinson’s and Huntington’s Diseases. In neurological disorders characterized by neuroinflammation, CB2R-based therapies could be an attractive alternative to targeting CB1R. In the case of cardiac and metabolic diseases, both central and peripheral CB1Rs are known to play a key role in their etiology. Central administration of cannabinoids into the PAG and RVLM triggers a sympathoexcitatory response, while administration in the NTS is known to improve baroreflex sensitivity and facilitates inhibition of pressor responses. Interestingly, CB1Rs were downregulated in the NTS of SHRs. However, myocardial CB1R was augmented in SHRs, and is implicated in mediating cardiodepressant effects. Stimulation of endothelial CB1R was also implicated in mediating vasodilation. Although some studies reported beneficial effects of CB1R agonism in hypertension, CB1R antagonism was demonstrated to improve both cardiovascular and endothelial function in metabolic syndrome. CB1R antagonism also improved functions of insulin-sensitive tissues such as liver and pancreas. Blockade of CB1R was shown to attenuate hepatocellular damage and beta cell loss, and also improve insulin signaling.

**Table 1 ijms-21-07693-t001:** Systematic review of recent * clinical studies reporting cannabinoid receptors as biomarkers, and/or investigating the efficacy of cannabinoids in pathological conditions.

Subjects and Measurement Indices	Study Design	CB receptors as Biomarkers	Outcome of Pharmacological Intervention	Reference
Measurement of CB1R density in brown adipose tissue in lean and obese healthy males	Non-randomized, crossover clinical trial	CB1R upregulation in obese individuals	CB1R blockade increased lipolysis	[186]
Determination of the effect of dronabinol in gut transit in irritable bowel syndrome with diarrhea.	double-blind, randomized, placebo-controlled, parallel-group study	*CNR1* rs806378 CT/TT is associated with a delay in colonic transit compared with CC	Dronabinol delays transit in individuals with *CNR1* rs806378 CT/TT	[187]
Impact of Sativex on *CNR1* and *CNR2* expression in peripheral blood mononuclear cells (PBMCs) in patients with multiple sclerosis (MS) secondary progressive (MSS-SP).	Controlled clinical trial	*-*	Significant decrease in *CNR2* expression	[188]
Measurement of CB1R density in the brains of schizophrenic individuals with or without antipsychotic medication	prospective study	Increased CB1R binding in mesocorticolimbic regions of individuals with schizophrenia	-	[189]
Measurement of CB1R density in the brains of pre- Huntington disease mutation carriers	prospective study	Decrease in CB1R density in prefrontal cortex compared to controls	-	[190]
Determination of whether hypocaloric diet and/or aerobic exercise alters subcutaneous adipose tissue CB1R and FAAH expression in obese women	Randomized clinical trial	Caloric restriction alone lowered gluteal CB1R and FAAH, while both caloric restriction plus aerobic exercise reduced abdominal adipose tissue FAAH gene expression	-	[191]
Determination of whether exercise training resulted in changes in muscle CB1R and TRPV1 expression in heart failure patients	Randomized controlled trial	Exercise training significantly increased gene expression of the TRPV1 receptor and the CB1R	-	[192]
Impact of single nucleotide polymorphism rs3123554 in CNR2 on metabolic and adiposity parameters in obese induvial on two hypocaloric diets	Randomized controlled trial	Individuals that are carriers for *CNR2* genetic variant loose less body weight.	-	[193]
Determination of whether *CNR2* gene variation rs35761398 (Q63R) is significantly associated with chronic idiopathic thrombocytopenic purpura (ITP) in children	Case-control association study	*CNR2* gene variation is significantly associated with childhood chronic ITP	-	[194]
Impact of interferon therapy on CB1R and CB2R gene expression in immune cells from patients on interferon therapy	Controlled trial	Reduction in both CB1R and CB2R after interferon therapy	-	[195]
Impact of Sativex on the clinical improvement of motor, cognitive and psychiatric measures in patients with Huntington’s Disease.	double-blind, randomized, cross-over, placebo-controlled, pilot trial	*-*	No improvement in motor, cognitive, and behavioral functional scores when compared to the placebo	[196]
Impact of pirfenidone on CB1 and CB2 gene expression in liver biopsies, in individuals with chronic hepatitis C	Open-label, non-controlled, and non-randomized clinical trial	*-*	Significant upregulation of *CNR2,* while no statistical difference in *CNR1.*	[197]
Determination of the effects of isocaloric low and high-fat diets on endocannabinoid system in obese individuals	randomized cross-over study	Reductions in skeletal muscle CB1R in high fat diet group	-	[198]
Measurement of VEGF and cytokines in sera of obese PCOS women in response to rimonabant.	Randomized open-labelled parallel study	-	CB1R blockade raises VEGF and the pro-inflammatory cytokine IL-8 in obese women with PCOS	[199]
Measurement of insulin-like growth factor I levels in women with anorexia nervosa (AN) in response to dronabinol	Prospective, double-blind randomized crossover study	-	Dronabinol affected neither the concentration nor the activity of the circulating IGF-system in women with severe and chronic AN.	[200]
Measurement of skin conductance response to determine whether dronabinol facilitates fear extinction learning in healthy individuals	Randomized, double-blind, placebo-controlled trial	-	Dronabinol facilitates extinction of conditioned fear in humans	[201]
Measurement of glucose-dependent insulinotropic polypeptide (GIP) and glucagon-like peptide-1 (GLP-1) responses during oral glucose tolerance test (OGTT) in the plasma of lean and obese participants	Randomized, double-blind, crossover, placebo-controlled trial	-	CBR agonist increased circulating GIP levels, but not GLP levels, in the fasting (nonstimulated) state.	[202]
Measurement of body weight and HbA_1c_ in obese and overweight individuals with and without diabetes in response to CB1R antagonist, CP-945,598.	double-blind, placebo-controlled trial	-	CP-945,598 resulted in a reduction in body weight and better glycemic control.	[203]
Impact of smoked medicinal cannabis on appetite hormones ghrelin, leptin and PYY in individuals with HIV-associated neuropathic pain	double-blind cross-over, placebo-controlled trial	-	Medical cannabis resulted in significant increases in plasma levels of ghrelin and leptin, and decreases in PYY	[204]
Impact of rimonabant on improving the risk of cardiovascular death, myocardial infarction, or stroke in individuals with cardiovascular risk factors or prior cardiovascular events.	Randomized, double-blind, placebo-controlled trial	-	Termination of trial due to neuropsychiatric effects	[205]
Impact of CB1R inverse agonist, taranabant, for maintenance of prior weight loss achieved on a low-calorie diet	Randomized, double-blind, placebo-controlled trial	-	Improvement in weight loss with taranabant, compared to maintenance therapy alone	[206]
Impact of CB1R inverse agonist, taranabant, on weight loss in obese and overweight patients	Randomized, double-blind, placebo-controlled trial	-	Taranabant treatment resulted in statistically significant weight loss	[207]
Impact of CB1 neutral antagonist tetrahydrocannabivarin on resting state functional connectivity in key brain regions relevant to development of obesity, in healthy individuals	Randomized, within-subject, double-blind, placebo-controlled trial	-	Alteration in resting state functional connectivity without significant effects on mood.	[208]
Impact of CB1 neutral antagonist tetrahydrocannabivarin on activation of brain regions involved in food aversion in healthy individuals	Randomized, double-blind, placebo-controlled trial	-	Treatment increased neural responses to aversive stimuli	[209]
Impact of CB1/CB2 receptor agonist KN38-7271 on survival rates following head injury	Randomized, double-blind, placebo-controlled phase II trial	-	Improved survival rates in the early phase of the comatose patient after a head injury	[210]
Impact of peripherally acting CB1/CB2 receptor agonist, AZD 1940, on capsaicin-evoked pain and hyperalgesia	Randomized, double-blind, placebo-controlled	-	No evidence of analgesic efficacy in the human capsaicin pain model.	[211]
Impact of oral dronabinol on the progression of primary and secondary progressive multiple sclerosis in patients with prior diagnosis	Randomized, double-blind, placebo-controlled study	-	No significant effect on the progression of multiple sclerosis in the progressive phase	[212]
Impact of oral dronabinol on altering pain threshold in individuals diagnosed with functional chest pain (FCP)	Randomized, double-blind, placebo-controlled study	-	Pain threshold was increased, and the frequency and intensity of pain was reduced, in FCP	[213]
Impact of rimonabant on changes in liver fat in individuals with metabolic syndrome	Randomized, double-blind, placebo-controlled trial	-	Reduction in liver fat is proportional to weight loss	[214]
Impact of rimonabant on carotid atherosclerosis in obese individuals	Randomized, double-blind, placebo-controlled trial	-	No significant effect on the progression of atherosclerosis	[215]
Impact of rimonabant on body weight in obese patients with binge eating disorders	randomized, double-blind, placebo-controlled study	-	Reduction in body weight when compared to placebo group	[216]
Impact of rimonabant on neurocognitive impairments in individuals with schizophrenia	randomized, double-blind, placebo-controlled study	-	Improvement on a probabilistic learning task, with no improvement in global cognitive functioning	[217]
Impact of rimonabant on fatty acid and triglyceride metabolism and insulin sensitivity after controlling for metabolic effects of weight loss in obese women	Randomized controlled trial	-	Increased lipolysis and fatty acid oxidation without any effect on insulin sensitivity	[218]
Impact of rimonabant on insulin regulation of free fatty acid and glucose metabolism, after controlling for weight loss, in obese, metabolic syndrome individuals	randomized, double-blind, placebo-controlled study	-	Improvement in insulin regulation of free fatty acids and glucose metabolism was due to weight loss	[219]
Treatment of dementia-related neuropsychiatric symptoms (NPS) in response to low-dose oral THC	Prospective study	-	low-dose THC does not significantly reduce NPS	[220]

* Clinical trials pertaining to the endocannabinoid system in cardiovascular, neurological and metabolic disorders from 2010–2020, are listed in this table.

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
