# Peer review of "Cannabinoid Receptors: An Update on Cell Signaling, Pathophysiological Roles and Therapeutic Opportunities in Neurological, Cardiovascular, and Inflammatory Diseases"

_ijms, 2020, doi:10.3390/ijms21207693_

Round 1
Reviewer 1 Report
I think that the Authors made a great job trying to make a "list" of all the roles of CB1R and CB2R, and the possible implications in pathology and therapeutics field.
However, the resulted work is too long and confusional, with a lot of examples and references to other literature. I can not review the Figures because they are not visible in the submission. I think that the work lacks of 2 important and more general paragraphs explaining the biological effects and the potential therapeutic effects, respectively: they will be very useful before to start to make a list of all the specific situations and pathologies. In this paragraphs Authors can review more recent literature, especially the one regarding the peripheral effects of cannabinoids, in peripheral tissues, like muscles, fascia and connective tissues. I think that it is very important and new, and it totally lacks in this work.
In general it is a good work that goes deeper in a lot of pathologies in a very specific way, but it needs a major revision to become more useful and interesting.
In details:
- paragraph 2.1: too long and detailed
- explain better the role of cannabinoids in nociception, regulation of inflammation, myofascial pain, pain reduction.
- lines 186-188: and Hu-308? and others?
- lines 218-219: and the peripheral tissues?
- paragraph 2.4.1: too long. A picture could be useful here.
- line 294: mu?
- crosstalk: too long
- 392 "intracellular signaling": it could be useful upper in the text
- 3.1.1: so both CB1R and CB2R are involved?
- 3.2.1: maybe Authors can add a picture
- line 561: BBB?
- line 701: LTD?
- references 393-400: there is something else more new in the recent literature
- line 948: the function of CBR as antinflammatory can be deepened
- line 1034: and in normal peripheral tissues?
Author Response
We would like to thank Reviewer 1 for their detailed review of our manuscript. We have carefully reviewed the comments and have endeavored to address the concerns of the reviewer. We have addressed the comments as follows:
- I think that the Authors made a great job trying to make a "list" of all the roles of CB1R and CB2R, and the possible implications in pathology and therapeutics field.
However, the resulted work is too long and confusional, with a lot of examples and references to other literature. I can not review the Figures because they are not visible in the submission. I think that the work lacks of 2 important and more general paragraphs explaining the biological effects and the potential therapeutic effects, respectively: they will be very useful before to start to make a list of all the specific situations and pathologies. In this paragraphs Authors can review more recent literature, especially the one regarding the peripheral effects of cannabinoids, in peripheral tissues, like muscles, fascia and connective tissues. I think that it is very important and new, and it totally lacks in this work.
Response: We thank the reviewer for their extremely helpful suggestions. Two new paragraphs have been included in the text for better readability. One at the beginning of section 3 (page 11 lines 20-25) to help connect the previous sections, which focusses on molecular and cellular roles of cannabinoid receptors, to their physiological and pathological roles, which is described in this section. The second paragraph is included under the immunomodulation section, 3.5.2, titled ‘pain’ (page 26 lines 1-11). In this paragraph we further describe the role of peripheral CB1R in various pain states. We believe that the peripheral effects of cannabinoids have been described in great detail under the metabolic, cardiovascular and also immunomodulation sections. For instance, we describe the role of peripheral CB1R in insulin-sensitive tissues such as skeletal muscle and adipocytes (3.3.2), bone and synoviocytes (3.5.2) to name a few.
- paragraph 2.1: too long and detailed
Response: We do feel that the detail is necessary in describing the various studies that have contributed immensely to our understanding of CB1R and CB2R structure and signaling. This provides the necessary framework for the section on signaling that comes later.
- explain better the role of cannabinoids in nociception, regulation of inflammation, myofascial pain, pain reduction.
Response: Thank you for this suggestion. We have an additional paragraph on pain under section 3.5.2 (page 26 lines 1-11). With regards to inflammation, we provided extensive evidence for the role of both CB1R and CB2R in neuroinflammation (3.2.2), and immunosuppression (3.5). We have also listed other reviews that have focused solely on CB1R and/or CB2R in the regulation of inflammatory responses.
- lines 186-188: and Hu-308? and others?
Response: Thank you for bringing this to our attention. We have included 4 more examples of synthetic cannabinoids in lines 186-188.
- lines 218-219: and the peripheral tissues?
Response: The distribution of CB1R in peripheral tissues is included in the following section titled ‘non-neuronal cells’( see page 6).
- paragraph 2.4.1: too long. A picture could be useful here.
Response: We apologize for the figures not being visible. Our initial submission did include a figure detailing both canonical and non-canonical CB1R signaling.
- line 294: mu?
Response: Thank you for this. Correction has been made- mu to m page 8
- crosstalk: too long
Response: Thank you for the suggestion. We hope that the figure for this section helps in distilling the most important points.
- 392 "intracellular signaling": it could be useful upper in the text
Response: This has been described in the figure as well. Please let us know if any more changes need to be made to this section. Apologies, but we are not sure what ‘upper’ means.
- 1.1: so both CB1R and CB2R are involved?
Response: While it is not surprising that CB1R is involved in embryogenesis, considering their roles in cell specialization and differentiation, there were a few studies that reported the role of CB2R in some facets of embryogenesis as well. We thought the readers may find this of interest and decided to include this in our review as well.
- 2.1: maybe Authors can add a picture
Response: Thank you for this suggestion. The sections on neuromodulation and neuroinflammation are shown in figure 4. We did not focus on LTP and LTD in our figure, since these two mechanisms have been described in detail in multiple reviews, and we believe that our review would not be adding anything new (Heifets and Castillo (2009)). The aspect on the interaction between CB1R-mediated neuroinflammatory responses and neuromodulation however has not been described in much detail.
- line 561: BBB?
Response: Thank you for this. Correction has been made for this abbreviation on page 15.
- line 701: LTD?
Response: The abbreviation was listed earlier in section 3.2.1 in line 505 page 13-long term synaptic depression (LTD).
- references 393-400: there is something else more new in the recent literature
Response: Thank you for this suggestion. We have listed two more recently conducted retrospective analysis for cannabinoids on pain see references 555, and 556 in section 3.5.2 page 26.
- line 948: the function of CBR as antinflammatory can be deepened
Response: Thank you for this. In this section, we focused on the anti-inflammatory potential with respect to only Rheumatoid arthritis. In the section in 3.2.2 and 3.5.1, we go into great detail on the role of not only CB2R, but also CB1R in modulating inflammatory states.
- line 1034: and in normal peripheral tissues?
Response: Thank you for this. Correction has been made page 27.
Reviewer 2 Report
The authors have made an extensive and well structured review on canabinoid receptors. I find the manuscript suitable for publication in the present form.
Additional comments:
I found the review quite extensive and that is why I was willing to accept it in the present form.
Nonetheless, this extensiveness might also be something that has to be corrected as it would not be read by the average reader completely.
Author Response
We would like to thank Reviewer 2 for their thoughtful review and comments on our manuscript. We hope we have addressed the concerns of the reviewer. We have addressed the comments as follows:
- The authors have made an extensive and well structured review on canabinoid receptors. I find the manuscript suitable for publication in the present form.
Additional comments:
I found the review quite extensive and that is why I was willing to accept it in the present form.
Nonetheless, this extensiveness might also be something that has to be corrected as it would not be read by the average reader completely.
Response: Thank you for this. We appreciate the kind words. We think that the readers would be find specific parts of this review to be more relatable than others since we have listed the utility of cannabinoid therapeutics in multiple pathological conditions. For instance, a researcher working on cardiac cells may find the section on hypertension more useful than the sections on anxiety. However, the pathological conditions we did list, we went into as much detail as possible, since those sections would most likely be useful to researchers working in that field.
Reviewer 3 Report
Very well written.
The only thing you could consider is line 122/123, explaining the word "peripheral". Later in the text you explain this, but it would be good to understand already here the difference in CB1 and CB2
Author Response
We would like to thank Reviewer 3 for their thoughtful review and comments on our manuscript as well. We hope we have addressed the concerns of the reviewer. We have addressed the comments as follows:
- Very well written.
The only thing you could consider is line 122/123, explaining the word "peripheral". Later in the text you explain this, but it would be good to understand already here the difference in CB1 and CB2
Response: Thank you for this. We appreciate the kind words. We have made the correction in lines 122/123 page 4.
Round 2
Reviewer 1 Report
Thank you for the deep work of revision.
The new version is more readable and I have received from the editor the Figures.
This new version can be accepted, but I think that still one point is missing: the CB1 and CB2 receptors are found also in the connective tissue (see for example Fede et al, Expression of the endocannabinoid receptors in human fascial tissue. Eur J Histochem.) This point can be deepened in Figure 2 and in the text. It's important to understand the peripheral effects of therapeutic applications of cannabinoids.
Author Response
We would like to thank Reviewer 1 for their detailed review of our manuscript. We have addressed the reviewer’s comments as follows:
Thank you for the deep work of revision.
The new version is more readable and I have received from the editor the Figures.
This new version can be accepted, but I think that still one point is missing: the CB1 and CB2 receptors are found also in the connective tissue (see for example Fede et al, Expression of the endocannabinoid receptors in human fascial tissue. Eur J Histochem.) This point can be deepened in Figure 2 and in the text. It's important to understand the peripheral effects of therapeutic applications of cannabinoids.
Response: We thank the reviewer for these suggestions. We added the following sentence in Section 2.3 Page 7 lines 11 -12: “Additionally both CB1R and CB2R have been detected in connective tissues such as fascial fibroblasts and osteoclast cells [108,109].” We also added to Figure 2 information depicting connective tissue.